# Decoding the physical principles of two-component biomolecular phase separation

Yaojun Zhang[1], Bin Xu[2], Benjamin G Weiner[2], Yigal Meir[2,3,4], Ned S Wingreen[4,5]*

[1]Center for the Physics of Biological Function, Princeton University, Princeton, United States; [2]Department of Physics, Princeton University, Princeton, United States; [3]Department of Physics, Ben Gurion University of the Negev, Beersheba, Israel; [4]Department of Molecular Biology, Princeton University, Princeton, United States; [5]Lewis-Sigler Institute for Integrative Genomics, Princeton University, Princeton, United States

**Abstract** Cells possess a multiplicity of non-membrane-bound compartments, which form via liquid-liquid phase separation. These condensates assemble and dissolve as needed to enable central cellular functions. One important class of condensates is those composed of two associating polymer species that form one-to-one specific bonds. What are the physical principles that underlie phase separation in such systems? To address this question, we employed coarse-grained molecular dynamics simulations to examine how the phase boundaries depend on polymer valence, stoichiometry, and binding strength. We discovered a striking phenomenon – for sufficiently strong binding, phase separation is suppressed at rational polymer stoichiometries, which we termed the magic-ratio effect. We further developed an analytical dimer-gel theory that confirmed the magic-ratio effect and disentangled the individual roles of polymer properties in shaping the phase diagram. Our work provides new insights into the factors controlling the phase diagrams of biomolecular condensates, with implications for natural and synthetic systems.

*For correspondence:
wingreen@princeton.edu

## Introduction

Eukaryotic cells are host to a multiplicity of non-membrane-bound compartments. Recent studies have shown that these compartments form via liquid-liquid phase separation (*Brangwynne et al., 2009*; *Li et al., 2012*; *Molliex et al., 2015*). The phase-separated condensates enable many central cellular functions – from ribosome assembly, to RNA regulation and storage, to signaling and metabolism (*Shin and Brangwynne, 2017*; *Banani et al., 2017*). Unlike conventional liquid-liquid phase separation, for example water-oil demixing, the underlying interactions that drive biomolecular phase separation typically involve strong one-to-one saturable interactions, often among multiple components (*Ditlev et al., 2018*). As a result, the phase diagrams of biomolecular condensates are complex and are sensitive to a variety of physical properties of the biomolecules, included number of binding sites, binding strengths, and additional nonspecific interactions. Importantly, these physical parameters can be subject to biological regulation, and can thus directly impact the organization and function of the condensates. It is therefore crucial to understand how the physical properties of the components shape the phase diagram of biomolecular condensates.

Biomolecular condensates typically contain tens to hundreds of types of molecules. Yet, when characterized in detail, only a small number of components are responsible for condensate formation (*Ditlev et al., 2018*). One class of such condensates are those formed by the association of two essential components. In the simplest case, each component consists of repeated domains/stickers that bind in a one-to-one fashion with the domains of the other component (*Figure 1A and B*;

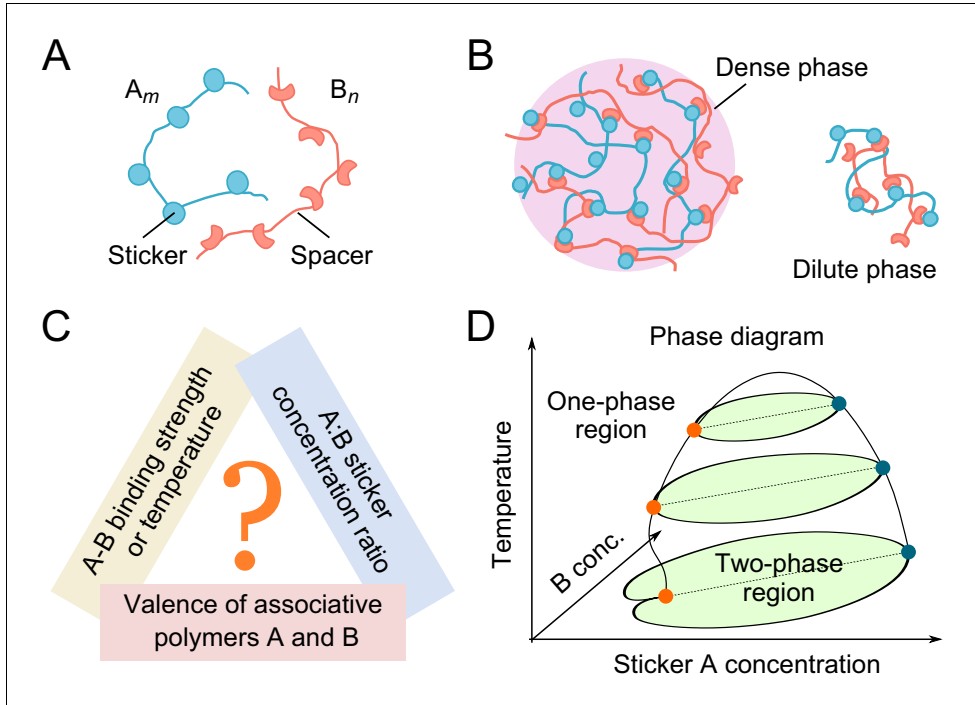

**Figure 1.** Phase behavior of sticker and spacer associative polymers. (**A**) Schematic of multivalent associative polymers. Each polymer consists of complementary domains (stickers) connected by flexible linkers (spacers). A and B denote the polymer type and $m$ and $n$ denote their valences (number of stickers). (**B**) Association of stickers drives phase separation, leading to the formation of a dense, network phase coexisting with a dilute phase of small oligomers (depicted by a dimer). (**C**) The phase diagram depends on variety of biologically tunable parameters. In this study, we focus on the effects of sticker-sticker binding strength, sticker:sticker concentration ratio (i.e. stoichiometry), and polymer valences. (**D**) Schematic of a representative 3D phase diagram of an $A_n : B_n$ system as a function of temperature (inverse of binding strength) and A and B sticker concentrations. The dilute-phase concentration displays anomalous dependence on the binding strength and sticker concentrations in the strong binding regime. This is the 'magic-ratio' effect which we explore here in detail.

*Choi et al., 2019*; *Xu et al., 2020*). Such two-component condensates have been observed in both natural and engineered contexts. For example, the pyrenoid, an organelle responsible for carbon fixation in the alga *Chlamydomonas reinhardtii*, is a condensate of the $CO_2$-fixing enzyme Rubisco with the linker protein Essential PYrenoid Component 1 (EPYC1). EPYC1 consists of five evenly-spaced Rubisco-binding regions, while Rubisco holoenzyme has eight specific binding sites for EPYC1. Multivalent interactions between Rubisco and EPYC1 are responsible for pyrenoid formation (*Freeman Rosenzweig et al., 2017*; *Wunder et al., 2018*; *He et al., 2020*). Promyelocytic leukemia (PML) nuclear bodies are condensates of PML proteins. PML is SUMOylated at three main positions and several minor sites. These modifications and a C-terminal SUMO Interaction Motif (SIM) found in most PML isoforms contribute to the formation of these bodies (*Shen et al., 2006*). Engineered polySUMO and polySIM proteins (10 repeats of Small Ubiquitin-like Modifier [SUMO] and SIM, respectively) phase separate when mixed together, but not as individual components (*Banani et al., 2016*; *Ditlev et al., 2018*).

Previous simulations (*Freeman Rosenzweig et al., 2017*; *Xu et al., 2020*) of average cluster size in such two-component systems revealed a striking phenomenon – for sufficiently strong binding, the formation of large clusters is suppressed when the valence of one species equals or is an integral multiple of the valence of the other species, favoring the formation of small stable oligomers instead of a condensate. The phenomenon reminiscent of the exact filling of atomic shells leading to the unreactive noble gases was termed the 'magic-number' effect. A similar effect was found in a ternary system modeling the clustering of nephrin, Nck, and NWASP proteins which regulates cell-cell adhesion in podocyte cells of the kidney (*Chattaraj et al., 2019*). However, cluster size may reflect a sol-gel percolation transition rather than a thermodynamic phase transition (*Harmon et al., 2017*), and

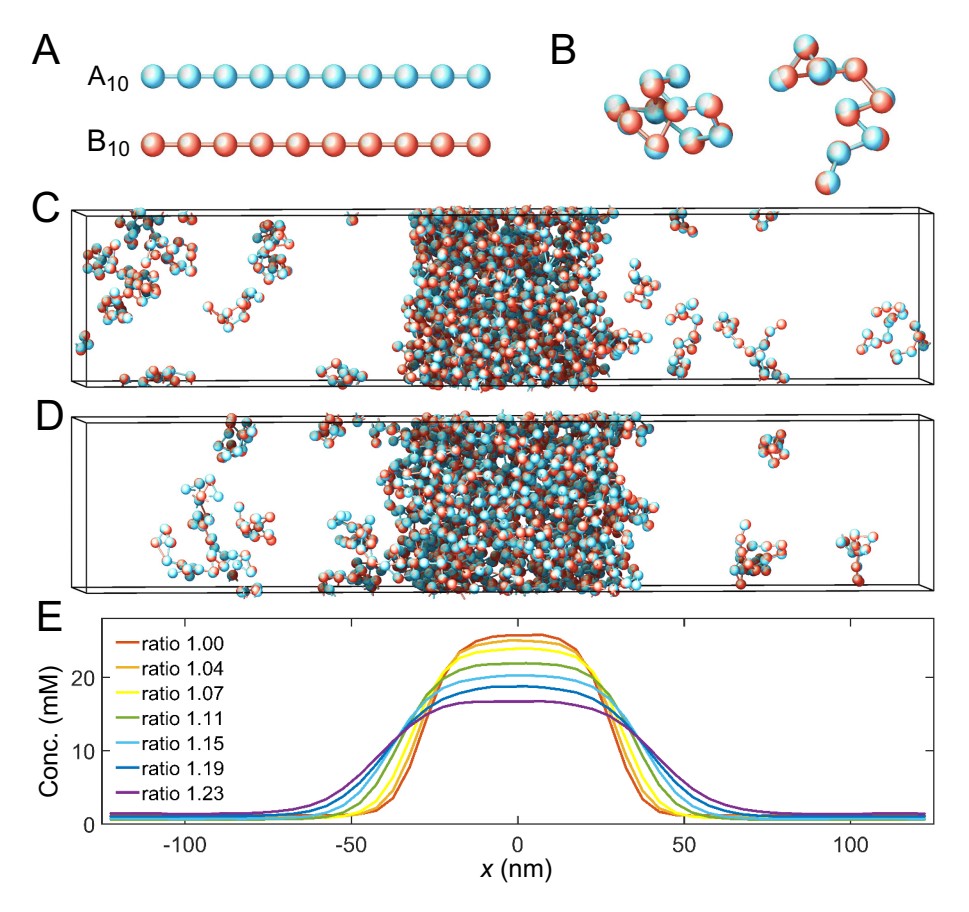

**Figure 2.** Coarse-grained molecular-dynamics simulations of two-component multivalent associative polymers. (**A**) The system consists of two types of polymers A (blue) and B (red) of varying lengths and concentrations. Depicted are A and B polymers of length 10, denoted as $A_{10}$ and $B_{10}$. Each polymer is modeled as a linear chain of spherical particles connected by harmonic bonds. Stickers of different types interact pairwisely through an attractive potential, while repulsion between stickers of the same type prevents them from overlapping and thus ensures one-to-one binding of stickers of different types (see Appendix 1 for details). (**B**) Snapshots of dimers formed by $A_{10}$ and $B_{10}$ with one-to-one bonds. (**C**) Snapshot of a simulation with 125 $A_{10}$ and 125 $B_{10}$ polymers. The system phase separates into a dense phase (middle region) and a dilute phase (two sides) in a 250 nm×50 nm×50 nm simulation box with periodic boundary conditions. (**D**) Same as *C* but with 138 $A_{10}$ and 112 $B_{10}$ polymers, yielding an overall sticker concentration ratio 1.23. (**E**) Sticker concentration profiles of $A_{10}$:$B_{10}$ systems at various overall sticker stoichiometries (total global sticker concentration fixed at 6.64 mM), each with the center of the dense phase aligned at $x = 0$ and averaged over time and over ten simulation repeats (see Appendix 1). All simulations performed in LAMMPS (*Plimpton, 1995*).

thus provides at best a qualitative measure of phase separation. Moreover, these previous studies focused on equal sticker stoichiometry, whereas biomolecular condensates cover a broad range of stoichiometries both in vitro (*Li et al., 2012*; *Banani et al., 2016*) and in vivo (*Sanders et al., 2020*).

Here, we directly delineate the full phase diagram of such two-component systems. Using coarse-grained molecular dynamics simulations, we explore systematically how phase boundaries depend on valence, stoichiometry, and binding strength of two associating polymers (*Figure 1C and D*). Our studies reveal an unanticipated effect – when the numbers of polymers of the two types have a rational stoichiometry (1:1, 1:2, etc.), phase separation can be strongly suppressed, which we call the 'magic-ratio' effect (*Figure 1D*, phase diagram at low temperatures). To understand the magic-ratio effects better, we develop a two-component sticker theory à la Semenov and Rubinstein (*Semenov and Rubinstein, 1998*). We model the system as dominated by polymer dimers in the dilute phase and by a condensate of independent stickers in the dense phase (*Figure 1B*). The

resulting analytical theory captures the magic-ratio effect discovered in simulations, and allows us to disentangle the individual roles of valence, stoichiometry, specific-bond strength, and nonspecific attraction in determining the phase boundaries of two-component multivalent systems. Living cells regulate the valence and interactions of biomolecules through chemical modification, or on a slower timescale, tune the stoichiometry via synthesis/degradation or sequestration, and over evolutionary time, adapt the strength of specific and nonspecific interactions through mutation of molecular sequences. Understanding the individual roles of these biologically tunable variables thus brings new insights into possible cellular strategies for regulating the formation and dissolution of biomolecular condensates.

## Results

### Coarse-grained molecular-dynamics simulations

We perform coarse-grained molecular-dynamics simulations using LAMMPS (*Plimpton, 1995*) to determine the phase boundaries of two-component multivalent systems (*Figure 2*). Briefly, we model the two polymer species as flexible linear chains of beads connected by harmonic springs (*Figure 2A*). Each bead represents one associative domain/sticker of the polymer. To ensure associative domains of different polymer types bind in a one-to-one fashion, we impose a finite-ranged attractive interaction between beads of different types. This, however, could lead to more than one-to-one associations. Therefore, to avoid such unwanted associations, we impose strong repulsive interactions between beads of the same type over a large enough range to prevent other beads overlapping with a bound pair, thus preventing multiple-to-one binding (*Figure 2B* and *Appendix 1—figure 1*), see Appendix 1 for details.

To find the binodal phase boundaries, we simulate hundreds of polymers of types A and B with, respectively, $m$ and $n$ stickers (an $A_m : B_n$ system) in a box with periodic boundary conditions (*Figure 2C and D*). We initialize the system by constructing a dense slab of polymers in the middle of the box (*Dignon et al., 2018*). The system evolves and relaxes according to Langevin dynamics (*Langevin, 1908*). After the system has achieved equilibrium, two phases coexist: a dilute phase consisting of dimers and other small oligomers, and a dense phase of an interconnected polymer condensate. We measure the corresponding density profile (*Figure 2E*) and calculate the dilute- and dense-phase concentrations by averaging the density profile over the regions ($x \leq -100\,\text{nm}$ or $x \geq 100\,\text{nm}$) and ($-10\text{nm} \leq x \leq 10\,\text{nm}$), respectively. See Appendix 1 for simulation details.

### Effect of valence

It was shown previously that for equal sticker stoichiometry in the strong-binding regime, clustering is substantially suppressed when the number of binding sites on one polymer species is an integer multiple of the number of binding sites on the other, as this condition favors the assembly of small oligomers in which all binding sites are saturated (*Freeman Rosenzweig et al., 2017*; *Xu et al., 2020*). What does this magic-number effect imply for the actual phase diagram? To address this question, we fix the valence of polymer A at 14 and systematically vary the valence of polymer B from 5 to 16 while keeping the two sticker concentrations the same, that is, at equal global sticker stoichiometry.

*Figure 3A and B* show simulation results for the total sticker concentrations of the dilute and dense phases for $A_{14}:B_5$ to $A_{14}:B_{16}$ systems. In the strong binding regime, for magic-number cases, that is when the valence of B is 7 or 14, the dilute-phase concentration shows pronounced peaks (*Figure 3A*, black curve). What is the origin of the peak at $A_{14}:B_{14}$? Intuitively, when the dilute phase of the two-component system is dominated by dimers (for systems $A_{14}:B_{12}$ to $A_{14}:B_{16}$, as supported by cluster size analysis in *Appendix 1—figure 2*), each of these dimers has high translational entropy, whereas polymers in the dense condensate have low translational entropy. For $A_{14}:B_{14}$, all binding sites can pair up in a dimer just as well as in the condensate, so the energy per polymer is not necessarily lower in the condensate. Why then is the condensate still competitive with the dilute phase? In a dimer, the binding sites of $A_{14}$ must match all the binding sites of $B_{14}$, leading to a reduced overall conformational entropy. By comparison, the polymers in the condensate are more independent, binding to multiple members of the other species and enjoying a relatively higher overall conformational entropy. Because the translational entropy of each dimer decreases as their

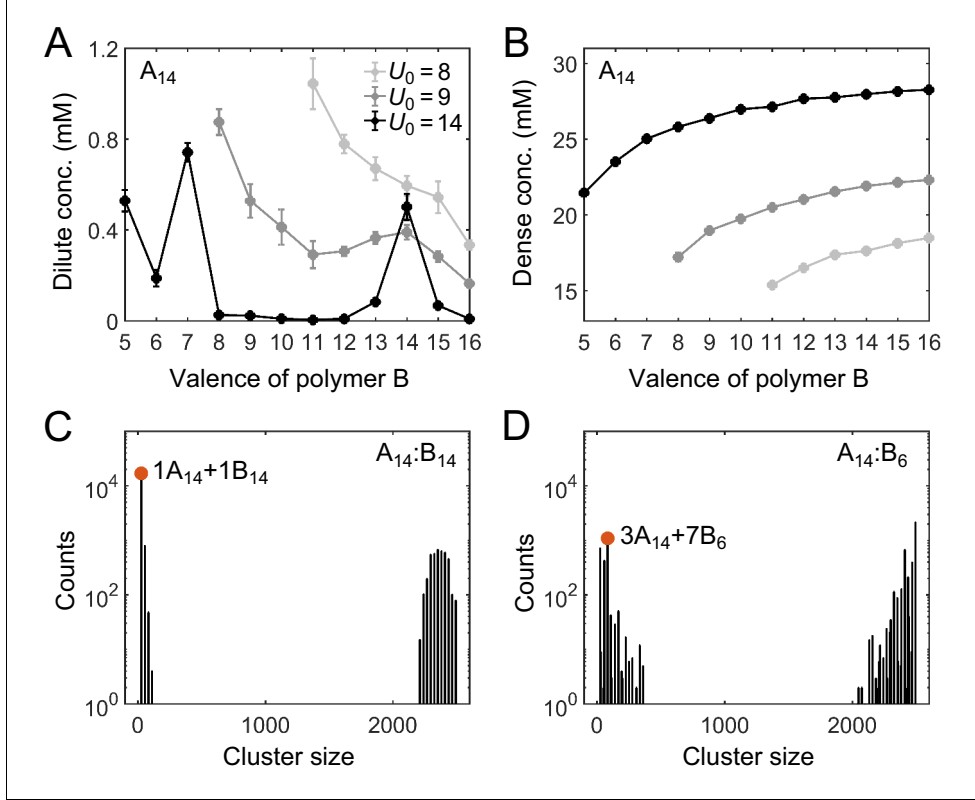

**Figure 3.** Simulations of associative polymers reveal a magic-number effect with respect to relative valence. Total sticker concentrations (type A plus type B) in (A) dilute and (B) dense phases for simulated polymer systems at different binding strengths. $U_0$ denotes the depth of the potential well, in units of $k_B T$ (see Appendix 1 for details). The valence of polymer A is 14, and the valence of polymer B ranges from 5 to 16. Global sticker stoichiometry is one and total global sticker concentration is 6.64 mM. Histograms of cluster size in (C) $A_{14}:B_{14}$ and (D) $A_{14}:B_6$ systems, for $U_0 = 14$. 'Counts' refer to number of clusters. Cluster size is measured in stickers. Red dots indicate the dominant oligomer in the dilute phase.

concentration goes up, the condensed phase eventually becomes more favorable and so the system phase separates with increasing concentration. Therefore, phase separation in $A_{14}:B_{14}$ is primarily driven by a competition between translational entropy and conformational entropy.

By contrast, for $A_{14}:B_{13}$ and $A_{14}:B_{15}$, one of the stickers in the dimer cannot be paired, and for $A_{14}:B_{12}$ and $A_{14}:B_{16}$, two stickers per dimer cannot be paired. Therefore, forming a condensate not only increases the conformational entropy but more importantly lowers the energy of these systems. This significantly tilts the balance in favor of condensation. As a result, the dilute-phase concentration is sharply peaked at $A_{14}:B_{14}$, falling off rapidly for increasingly unequal polymer lengths. We note that the dense-phase concentration shows no such feature (*Figure 3B*), indicating that the peak at $A_{14}:B_{14}$ does not arise from differences in the internal structure of the dense phase.

The dilute phase of two-component systems is not always dominated by dimers (*Appendix 1— figure 2*). For example, the dilute phase of the $A_{14}:B_7$ system is dominated by fully-bonded trimers with 1 $A_{14}$ and 2 $B_7$, the dilute phase of $A_{14}:B_8$ is dominated by trimers with 1 $A_{14}$ and 2 $B_8$, which has two unpaired stickers per trimer, and the dilute phase of $A_{14}:B_6$ is dominated by oligomers with 3 $A_{14}$ and 7 $B_6$, which although fully-bonded is not small (*Figure 3D*). Consistent with the above logic, we find another peak in the dilute-phase concentration at $A_{14}:B_7$ (*Figure 3A*). More generally, in contrast to the magic-number systems, the dilute phases in other cases are dominated by oligomers which are not capable of being fully bonded (high energy) and/or not small (low translational entropy) (*Appendix 1—figure 2*). The dilute-phase concentration is therefore lower in these non-magic-number cases.

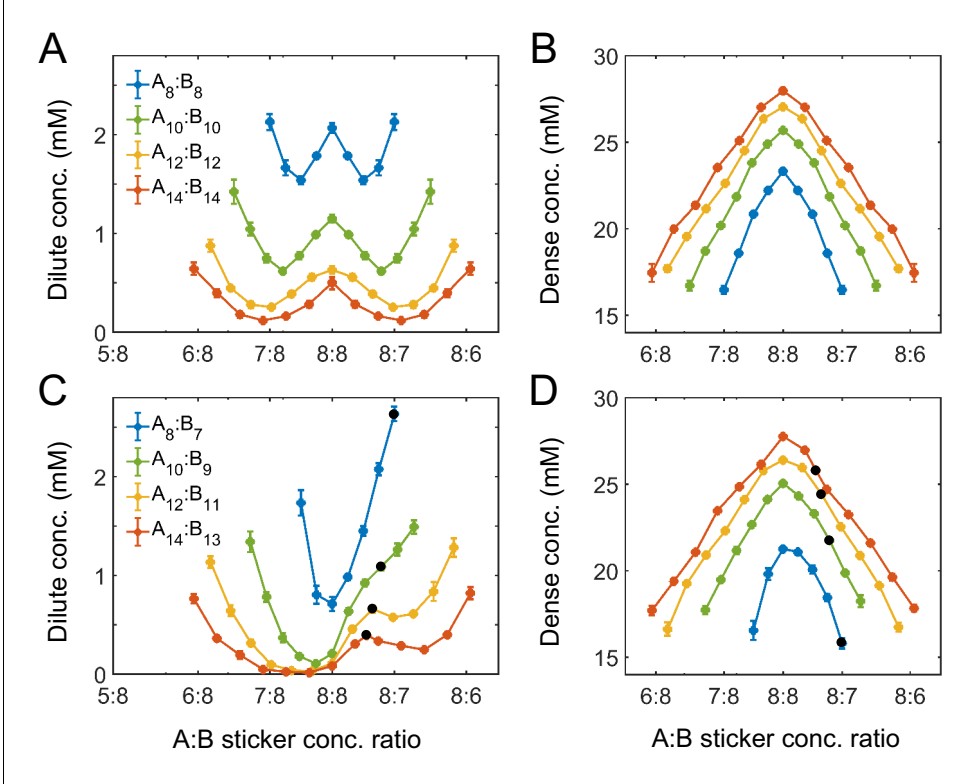

**Figure 4.** Simulations of associative polymers reveal a magic-ratio effect with respect to polymer stoichiometry. Sticker concentrations in (**A**) dilute and (**B**) dense phases for equal polymer length systems (i.e. $A_n : B_n$) at different global sticker stoichiometries. Sticker concentrations in (**C**) dilute and (**D**) dense phases for systems where polymer B is one sticker shorter than polymer A (i.e. $A_n : B_{n-1}$) at different global sticker stoichiometries; black dots indicate cases where the number of polymers of each type is the same. Interaction strength $U_0 = 14$ and total global sticker concentration 6.64 mM.

## Effect of binding strength

How do the phase boundaries depend on the strength of binding? *Figure 3A* shows that, for non-magic-number systems, the dilute-phase concentration decreases monotonically with increasing binding strength, whereas for magic-number systems the dependence can be non-monotonic. This difference is attributed to the distinct underlying driving forces for phase separation. For non-magic-number systems, as clustering allows a larger fraction of binding sites to be paired, the stronger the binding, the more the energy is lowered by condensate formation. Therefore, the dilute-phase concentration drops as binding strength increases (or as temperature decreases). Such energy-dependence is expected for conventional phase-separation models, such as *Flory, 1942*; *Huggins, 1941*.

Interestingly, for the magic-number system $A_{14}:B_{14}$, the dilute-phase concentration first decreases with increasing binding strength in the weak binding regime, similar to non-magic-number systems. However, as the binding energy is increased further, most of binding sites pair up in both dilute and dense phases. Phase separation is then primarily driven by a competition between conformational and translational entropy. The pairing up of binding sites reduces the conformational entropy of both the dense and dilute phases. By contrast, the translational entropy of the dilute-phase components is almost unaffected. Consequently, the dilute phase becomes more competitive relative to the condensate, so the dilute phase boundary shifts to higher concentration.

By comparison, the dense-phase concentration increases monotonically with increasing binding strength for all systems (*Figure 3B*). This follows because the stronger the binding, the more stickers are paired, which tightens the condensate structure. We note that at substantially higher binding energies than studied here, essentially all the binding sites are satisfied in both magic-number and non-magic-number systems, and the phase boundaries become independent of binding energy.

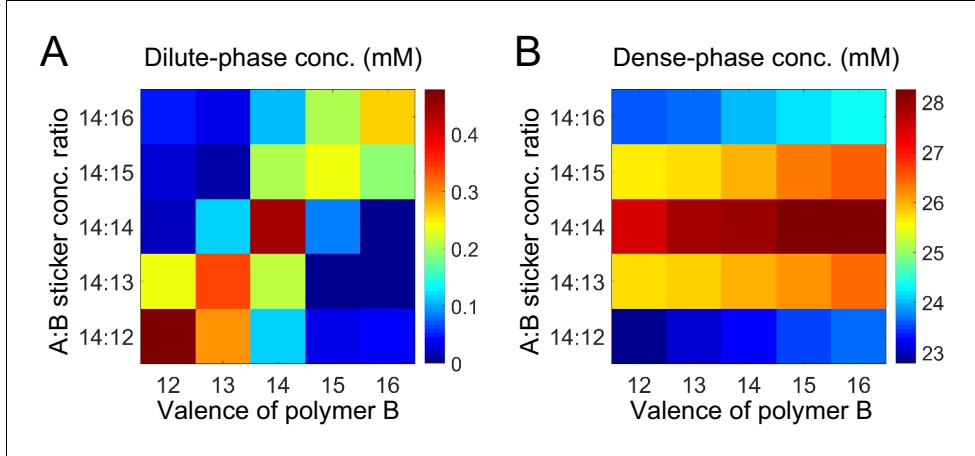

**Figure 5.** Simulations of associative polymers reveal a magic-ratio effect. Sum of concentrations of stickers A and B in (**A**) dilute and (**B**) dense phases for systems $A_{14}:B_{12-16}$ at global sticker stoichiometries 14:12-16. Parameters: interaction strength $U_0 = 14$ and total global sticker concentration 6.64 mM.

## Effect of sticker stoichiometry

How do the phase boundaries depend on overall sticker stoichiometry? *Figure 4A and B* show total sticker concentrations of the dilute and dense phases for magic-number systems $A_8:B_8$ to $A_{14}:B_{14}$ at different global sticker stoichiometries. For each system, the dilute-phase concentration peaks at equal sticker ratio, falls off initially as the ratio deviates from 1, and then curves back up. What is the origin of the peak at equal sticker stoichiometry? Recall that, in the strong binding regime, phase separation of magic-number systems is primarily driven by a competition between translational entropy and conformational entropy. Now consider starting with a system at equal sticker concentration, and adding more of one polymer species to the system. At the beginning, the added polymers readily enter the dense phase, which relaxes the conformational constraint that every sticker in the condensate has to pair with a partner. This increase of the conformational entropy of the condensate makes it more competitive, so the dilute-phase concentration decreases. However, as the ratio between the two polymers is increased further, it becomes possible to form a spectrum of dilute-phase oligomers which typically contain one extra polymer of the majority type (*Appendix 1—table 1*). These new oligomers have more relaxed structures than fully bonded dimers, which raises the conformational entropy of the dilute phase. Therefore, the dilute phase is favored over the condensate and its concentration curves back up.

*Figure 4A* also reveals that the dilute-phase concentration decreases with increasing polymer valence. This follows in part because translational entropy in the dilute phase is per dimer center of mass, whereas conformational entropy in both phases scales with the number of stickers. The entropic gain of joining the dense phase is therefore more on a per sticker basis for longer polymers, so the dilute-phase concentration decreases with increasing valence. As a less apparent yet important point, *Figure 4A* also shows that increasing polymer valence enhances both the width and relative height of the peak in the dilute-phase concentration. The inferred phase diagram for the $A_8:B_8$ system at $U_0 = 14k_BT$ is shown in *Appendix 1—figure 3* together with the homogeneous gelation/percolation threshold obtained at $U_0 = 8k_BT$. We also report in *Appendix 1—figure 5C* the volume fraction of the polymers in the dense phase, which is ~10%, comparable to the volume fraction of proteins in the cell cytoplasm.

*Figure 4C and D* show total sticker concentrations of the dilute and dense phases for unequal valence polymers $A_8:B_7$ to $A_{14}:B_{13}$ at different global sticker stoichiometries. The dilute phase boundary shows a symmetric minimum around equal stoichiometry for $A_8:B_7$, yet surprisingly, the phase boundary becomes asymmetric and then peaks at equal *polymer* stoichiometry with increasing polymer length (*Figure 4C*). What is the origin of these peaks? Taking the $A_{14}:B_{13}$ system as an example, its dilute phase is dominated by dimers with an unpaired A sticker. This strongly disfavors the dilute phase in the strong binding regime at equal sticker stoichiometry. However, as the overall A:B sticker stoichiometry increases, the excess As cannot be paired anyway. In particular, at equal

polymer stoichiometry (denoted as black dots in *Figure 4C*), forming dimers is no longer energetically costly. Therefore, to the left of the $A_{14}:B_{13}$ peak at equal polymer stoichiometry, the dilute-phase concentration is low because dimers are energetically disfavored as more bonds can be satisfied in the condensate. By contrast, to the right of the peak, the dilute-phase concentration is low for a different reason – because the condensate is entropically favored, similar to the peak with respect to stoichiometry for magic-number systems. Eventually, the dilute-phase concentration curves back up due to formation of higher oligomers in the dilute phase, as discussed for magic-number systems.

We note that for all these systems the dense-phase concentration shows no such striking features. Rather, the concentration decreases monotonically as the global sticker stoichiometry departs from one and as the valence of polymers decreases (*Figure 4B and D*).

## Effect of valence and stoichiometry

Above, we considered the role of both relative valence and relative stoichiometry. By plotting phase boundaries as joint functions of valence and stoichiometry, we obtain a unified picture: *Figure 5A and B* show the dilute- and dense-phase concentrations for systems $A_{14}:B_{12-16}$ at global sticker stoichiometries 14:12-16. Notably, the dilute-phase concentration is peaked along the diagonal (*Figure 5A*), that is at equal *polymer* stoichiometry, which we term the 'magic-ratio' effect because it occurs for rational ratios of associative polymers. Intuitively, all cases along the diagonal favor 1:1 polymer dimers: the dimers enjoy high translation entropy and there is no energy penalty involved in their formation. Thus, a dilute phase of dimers is strongly favored at equal polymer stoichiometry.

As for the dense phase concentration, it decreases monotonically as the global sticker stoichiometry departs from one and as the valence of polymers decreases (*Figure 5B*). This again indicates that the anomalous dependence of the dilute-phase concentration on valence and stoichiometry does not arise from special properties of the dense phase.

## Dimer-gel theory

While our simulations have revealed that a magic-ratio effect influences the boundaries of phase separation for associating polymers, we desire a deeper understanding of the interplay of factors such as overall valence, stoichiometry, and interaction strength. To this end, we develop a mean field theory of two-component associative polymers à la Semenov and Rubinstein (*Semenov and Rubinstein, 1998*; *Xu, 2018*).

Specifically, we consider a system of A and B polymers as in our simulations. Each polymer is a linear chain of $L_1$ or $L_2$ stickers of type A or type B, respectively. Without loss of generality, we take $L_1 \geq L_2$. stickers of different types associate in a one-to-one fashion. Our simulations suggest that for polymers of similar valence close to equal polymer stoichiometry the dilute phase is dominated by dimers and the dense phase is a gel network. Therefore, we assume that polymers can associate either as dimers or, alternatively, as a condensate in which pairs of stickers bind independently. This assumption of independence is a mean field approximation, as it neglects correlations between stickers in the same chain, and thus only applies when the polymers strongly overlap, that is at densities above the semidilute regime (*De Gennes, 1979*).

The partition function of such a system can be divided into three parts: $Z = Z_{ni}Z_sZ_{ns}$, where $Z_{ni}$, the partition function of a solution of non-interacting polymers, captures the translational and conformational entropy of the two polymer species, $Z_s$ captures specific interactions between associating stickers, and $Z_{ns}$ captures all nonspecific interactions.

The corresponding free-energy density for the mixed non-interacting polymers is *Semenov and Rubinstein, 1998*:

$$\frac{F_{ni}}{k_B T} = \frac{c_1}{L_1}\ln\frac{c_1}{eL_1} + \frac{c_2}{L_2}\ln\frac{c_2}{eL_2}, \tag{1}$$

where $c_1$ and $c_2$ are the concentrations of A and B polymers measured in terms of stickers. Note that the terms for the conformational entropy of non-interacting polymers are omitted in *Equation 1*, as they are linear in $c_1$ and $c_2$ and thus do not influence the phase boundaries.

To include specific interactions, we first consider the partition function $Z_s(N_{d1}, N_{d2}, N_b)$ for states with exactly $N_{d1}$ and $N_{d2}$ total numbers of stickers of A and B types in dimers (i.e. number of dimers equals $N_{d1}/L_1 = N_{d2}/L_2$) and $N_b$ additional sticker pairs,

$$Z_s(N_{d1}, N_{d2}, N_b) = P(N_{d1}, N_{d2}, N_b) W(N_{d1}, N_{d2}, N_b) \exp(N_{d1}\epsilon_d/L_1 + N_b\epsilon_b). \tag{2}$$

In *Equation 2*, $P$ is the number of different ways that polymers and stickers can be chosen to pair up to form dimers and independent bonds,

$$P(N_{d1}, N_{d2}, N_b) = \binom{N_1/L_1}{N_{d1}/L_1}\binom{N_2/L_2}{N_{d2}/L_2}(N_{d1}/L_1)!\binom{N_1 - N_{d1}}{N_b}\binom{N_2 - N_{d2}}{N_b}N_b!, \tag{3}$$

where $N_1$ and $N_2$ are the total numbers of stickers of A and B types. (Note that in *Equation (3)* if $L_1 > L_2$, the excess stickers of type A in dimers do not form additional bonds.) In *Equation (2)*, $W$ is the probability that all chosen polymers and stickers are, respectively, close enough to their specified partners in the non-interacting state to form dimers and independent bonds,

$$W(N_d, N_b) = \left(\frac{v_d}{V}\right)^{\frac{N_{d1}}{L_1}}\left(\frac{v_b}{V}\right)^{N_b}, \tag{4}$$

where $v_d$ and $v_b$ are effective interaction volumes and $V$ is the system volume. The last term in *Equation (2)* is the Boltzmann factor for specific interactions, where $\epsilon_d$ and $\epsilon_b$ are the effective binding energies of dimers and sticker pairs, in units of $k_B T$.

The part of the free-energy density due to specific interactions is

$$\frac{F_s}{k_B T} = -\frac{1}{V}\ln Z_s. \tag{5}$$

Using Stirling's approximation $\ln N! = N\ln N - N$, we obtain

$$\frac{F_s}{k_B T} = -\frac{c_1}{L_1}\ln c_1 + (1 - L_1)\frac{c_1 - c_{d1}}{L_1}\ln(c_1 - c_{d1}) + (c_1 - c_{d1} - c_b)\ln(c_1 - c_{d1} - c_b)$$
$$-\frac{c_2}{L_2}\ln c_2 + (1 - L_2)\frac{c_2 - c_{d2}}{L_2}\ln(c_2 - c_{d2}) + (c_2 - c_{d2} - c_b)\ln(c_2 - c_{d2} - c_b) + \frac{c_{d1}}{L_1}\ln(ec_{d2}L_1 K_d) + c_b\ln(ec_b K_b), \tag{6}$$

where $K_d \equiv e^{-\epsilon_d}/v_d$ and $K_b \equiv e^{-\epsilon_b}/v_b$ are, respectively, the dissociation constants of a dimer and of a pair of stickers. $c_{d1}$ and $c_{d2}$ are the concentrations of stickers of A and B types in dimers (so $c_{d1}/L_1 = c_{d2}/L_2$), and $c_b$ is the concentration of independent bonds.

In the thermodynamic limit, $F_s$ will be minimized with respect to $c_{d1}$, $c_{d2}$ and $c_b$, which implies

$$K_d c_{d2} L_1 (c_1 - c_{d1})^{L_1 - 1}(c_2 - c_{d2})^{L_2 - 1} = (c_1 - c_{d1} - c_b)^{L_1}(c_2 - c_{d2} - c_b)^{L_2}, \tag{7}$$

$$K_b c_b = (c_1 - c_{d1} - c_b)(c_2 - c_{d2} - c_b). \tag{8}$$

Note that if $c_b$ in *Equation (7)* and $c_{d1}$ and $c_{d2}$ in *Equation (8)* are set to zero, these equations reduce to

$$K_d \rho_d = (\rho_1 - \rho_d)(\rho_2 - \rho_d), \tag{9}$$

$$K_b c_b = (c_1 - c_b)(c_2 - c_b), \tag{10}$$

where $\rho_1$, $\rho_2$, and $\rho_d$ are the total concentrations of A and B polymers and dimers (measured in polymeric units), that is, $\rho_1 = c_1/L_1$, $\rho_2 = c_2/L_2$, and $\rho_d = c_{d1}/L_1 = c_{d2}/L_2$. *Equations (9) and (10)* are consistent with the definitions of the dissociation constants of a dimer and of an independent bond, respectively.

The free-energy density due to nonspecific interactions can in general be written as a power expansion in the concentrations (*Semenov and Rubinstein, 1998*; *De Gennes, 1979*),

$$\frac{F_{ns}}{k_B T} = \frac{1}{2}\sum_{ij} v_{ij}c_i c_j + \frac{1}{6}\sum_{ijk} w_{ijk}c_i c_j c_k, \tag{11}$$

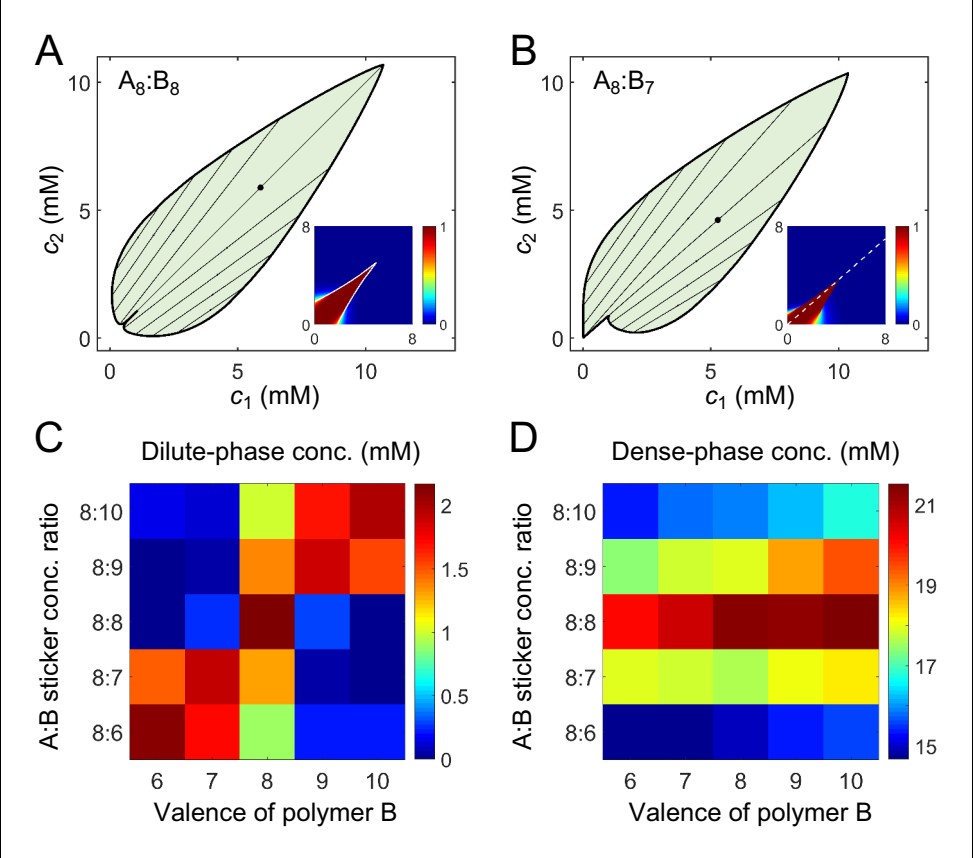

**Figure 6.** A dimer-gel theory predicts the magic-ratio effect. Phase diagrams of (**A**) $A_8:B_8$ and (**B**) $A_8:B_7$ systems: one-phase region white, two-phase region green. The dilute- and dense-phase concentrations are connected by representative tie lines. The tie line along the direction of equal polymer stoichiometry is denoted with a black dot. Insets: fraction of stickers in dimers for (**A**) $A_8:B_8$ and (**B**) $A_8:B_7$ systems. White curve in A inset is the transition boundary between dimer- and independent bonds-dominated regions predicted by $c_s$. Dashed white line in (**B**) inset denotes equal polymer stoichiometry. Sticker concentrations in (**C**) dilute and (**D**) dense phases for systems $A_8:B_{6\text{-}10}$ at global sticker stoichiometries 8:6-10. The total global sticker concentration is the same as in simulations, 6.64 mM. For details see Appendix 2. Parameters: $v_b = 9 \times 10^{-2} \text{mM}^{-1}$, $w_b = 7 \times 10^{-3} \text{mM}^{-2}$, $K_b = 3.8 \times 10^{-3} \text{mM}$, and $K_d$ values in **Appendix 1—table 3**.

where the sum is over all the species in the system, including free polymers/stickers, dimers and independent bonds, and $v_{ij}$ and $w_{ijk}$ are two- and three-body interaction parameters. For our simulation system, we derive a specific form of $F_{ns}$ by taking into account that (1) we are interested in the strong-binding regime where the magic-ratio effect is observed, (2) there is no nonspecific interaction between free polymers of different types in our simulation, and (3) nonspecific interactions are only important at high concentrations. The result is

$$\frac{F_{ns}}{k_B T} = \frac{v_b}{2}\max(c_1, c_2)^2 + \frac{w_b}{6}\max(c_1, c_2)^3, \tag{12}$$

where $v_b$ and $w_b$ are the two- and three-body interaction parameters for a solution of independent bonds. See Appendix 2 for details of the derivation.

Finally, substituting the conditions **Equations (7) and (8)** into **Equation (6)**, we obtain the total free-energy density $F = F_{ni} + F_s + F_{ns}$,

$$\frac{F}{k_B T} = \frac{c_1}{L_1}\ln\frac{c_1 - c_{d1}}{eL_1} + c_1\ln\frac{c_1 - c_{d1} - c_b}{c_1 - c_{d1}} + \frac{c_{d1}}{L_1}$$
$$+ \frac{c_2}{L_2}\ln\frac{c_2 - c_{d2}}{eL_2} + c_2\ln\frac{c_2 - c_{d2} - c_b}{c_2 - c_{d2}} + c_b + \frac{v_b}{2}\max(c_1, c_2)^2 + \frac{w_b}{6}\max(c_1, c_2)^3, \tag{13}$$

where $c_{d1}$, $c_{d2}$, and $c_b$ are the solutions of *Equations (7) and (8)*. *Equations (7), (8) and (13)* form a complete set which predicts the free-energy density of the two-component associative polymer system at given total global sticker concentrations, $c_1$ and $c_2$, of the two species.

Intuitively, in the strong-binding regime, that is when $c_1, c_2 \gg K_d, K_b$, polymers either associate as dimers or as independent bonds depending on their relative free energies. In the limit that dimers are preferred ($\rho_d = \min(\rho_1, \rho_2)$ and $c_b = 0$), the contribution from specific interactions is

$$\frac{F_s^{\text{dim}}}{k_B T} = \rho_d \ln K_d + (\rho - \rho_d) \ln \frac{\rho - \rho_d}{e} - \rho \ln \frac{\rho}{e}, \tag{14}$$

where $\rho = \max(\rho_1, \rho_2)$ is the concentration of the majority species in polymeric units. The terms on the right of *Equation (14)* reflect, respectively, the free-energy density due to dimer formation, translational entropy of leftover polymers, and loss of translational entropy of the majority species (in effect, the formation of each dimer removes the translation entropy of one free polymer). In the opposite limit that independent bonds are preferred ($c_b = \min(c_1, c_2)$ and $\rho_d = 0$),

$$\frac{F_s^{\text{ind}}}{k_B T} = c_b \ln K_b + (c - c_b) \ln \frac{c - c_b}{e} - c \ln \frac{c}{e}, \tag{15}$$

where $c = \max(c_1, c_2)$, and the terms are analogous to those in *Equation (14)*. Numerical studies show that the full $F_s(c_1, c_2)$ in *Equation (6)* is always well approximated by the lower of the two limiting values of $F_s$ (*Equation (14) and (15)*).

In which regions of concentration space are dimers versus independent bonds preferred? For a magic-number system composed of two polymer species of valence $L$ at equal sticker stoichiometry, $F_s^{\text{dim}}/k_B T = \rho \ln(K_d e/\rho)$ and $F_s^{\text{ind}}/k_B T = c \ln(K_b e/c)$. Comparing the two expressions, dimers are favored at low concentrations, whereas a network of independent bonds is favored at high concentrations. The transition occurs when $F_s^{\text{dim}} = F_s^{\text{ind}}$, that is at concentration $c_0 = e(K_b^L/(K_d L))^{1/(L-1)}$. Away from equal stoichiometry, the transition occurs at a lower concentration $c_s = c_0(s-1)^{s-1}s^{-s}$, where $s = \max(c_1, c_2)/\min(c_1, c_2) > 1$ (see Appendix 2 for details). As $c_s$ decreases rapidly with increasing $s$ (*Figure 6A* inset, white curve), the preference for dimers over a gel exhibits a sharp peak around equal stoichiometry.

To give a concrete example of the above analysis, we extract the values of $K_d$ for dimers from simulations, choose a value of $K_b$ for independent bonds close to the dissociation constant of a pair of stickers (see Appendix 2 for details), and numerically solve *Equation (7)* and (8) for $c_{d1}$, $c_{d2}$, and $c_b$ to find the fraction of stickers in dimers and independent bonds for all concentrations $(c_1, c_2)$. We find that indeed for polymers of equal valence, dimers are favored at low concentrations and independent bonds at high concentrations. The dimer dominated region extends sharply to higher concentrations in a narrow zone around the diagonal, as quantitatively captured by $c_s$ (*Figure 6A* inset and *Appendix 2—figure 1A*). For polymers of similar but unequal valence, the dimer dominated region extends to higher concentrations along the direction of equal *polymer* stoichiometry (*Figure 6B* inset and *Appendix 2—figure 1B*).

Finally, to extract the binodal phase boundaries, we substitute the values of $c_{d1}$, $c_{d2}$, and $c_b$ into *Equation (13)* to first obtain the free energy as a function of $c_1$ and $c_2$. The free-energy landscape has two basins, one at small concentrations corresponding to the dilute dimer-dominated phase, and one at high concentrations corresponding to the dense independent-bond-dominated gel-phase (*Appendix 2—figure 2*). We locate the phase boundaries by applying convex-hull analysis to this free-energy landscape (see Appendix 2).

Does the dimer-gel theory capture the magic-ratio effect revealed by our simulations? *Figure 6A and B* show the phase diagrams of $A_8$:$B_8$ and $A_8$:$B_7$ systems. In both cases, the phase boundaries on the dilute side extend sharply into the two-phase region along the direction of equal polymer stoichiometry (tie lines along this direction are denoted by black dots). *Figure 6C and D* show the dilute- and dense-phase concentrations for systems $A_8$:$B_{6-10}$ at global sticker stoichiometries 8:6-10. Notably, the dilute-phase concentrations are substantially shifted up around the diagonal, verifying the magic-ratio effect observed in simulations (*Figure 5A*).

One of the major assumptions of the dimer-gel theory is a mean-field approximation. Mean-field theory ignores correlations in binding between stickers in the same chain, and therefore has been applied to long chains in the weak binding regime (such that not every sticker is bound)

(*Prusty et al., 2018*; *Choi et al., 2020b*). Our dimer-gel theory bypasses this stringent requirement by explicitly assuming the dilute-phase components to be dimers, and only considers stickers to associate independently in the dense phase. This approximation captures a key feature of the dense phase, namely that a single polymer binds to multiple partners. Nevertheless, because stickers belonging to the same polymer are tethered together with relatively short linkers in our simulations, correlations in binding exist (*Appendix 2—figure 5A*). Therefore, what should be considered to be 'independent' is not individual stickers but rather segments of the binding correlation length (~1.8 stickers). The dense phase of a valence 14 system is thus more accurately described by the theory at valence $14/1.8 \approx 8$. We therefore present results for valence eight systems in *Figure 6*. (The theoretical phase diagrams and the dilute- and dense-phase concentrations for valence 14 systems also verify the magic-ratio effect (*Appendix 2—figure 4*)).

The dimer-gel theory has only a handful of parameters: the valences $L_1$ and $L_2$ of polymers A and B, the dissociation constants $K_d$ and $K_b$ of dimers and independent bonds, and the nonspecific interaction parameters $v_b$ and $w_b$. How are the phase boundaries and the magic-ratio effects determined collectively by these parameters? If valence is increased while keeping all other parameters fixed in the theory, for equal valence polymers we find that the dilute-phase concentration decreases, while the dense-phase concentration increases, and the peak with respect to stoichiometry is enhanced in terms of the dilute-phase peak-to-valley ratio (*Appendix 2—figure 6A and B*). If valence is increased for unequal valence polymers, we observe that the shape of the dilute phase boundary transitions from a shoulder to a peak (*Appendix 2—figure 6C and D*). All these features are consistent with the simulation results in *Figure 4*.

For the theory to agree quantitatively with the phase boundaries from simulations, we find that smaller values of nonspecific interaction parameters are necessary for higher valence systems (*Appendix 2—figure 6C and D*). Intuitively, this follows because higher valence polymers have more backbone bonds, which bring bound sticker pairs closer together in the dense phase – effectively reducing the nonspecific repulsion between them. Finally, the dimer-gel theory also predicts that the magic-ratio effect disappears in the weak-binding regime (*Appendix 2—figure 7*), consistent with our simulation results (*Figure 3*).

## Discussion

Intracellular phase separation is driven by multivalent interactions between macromolecules. These interactions are separated into two classes (*Ditlev et al., 2018*; *Pak et al., 2016*): (1) specific interactions, such as binding between protein domains, are relatively strong and involve specific partners and (2) nonspecific interactions, such as electrostatic and hydrophobic interactions, which are much weaker, more generic, and non-saturable. Multivalent systems with specific interactions allow for 'orthogonal' condensates to form: the specific interactions holding together one class of droplets will typically not interfere with those holding together another class. Motivated by the key role of specific interactions in intracellular phase separation, we focused on exploring the effects of specific interactions on the phase boundaries of two-component associative polymers. Specifically, we combined coarse-grained molecular dynamics simulations and analytical theory to examine the individual roles of valence, stoichiometry, and binding strength on the phase boundaries. In particular, we identified a magic-ratio effect: for sufficiently strong binding, phase separation is strongly suppressed at equal *polymer* stoichiometry.

The magic-ratio effect occurs exclusively in the strong-binding regime. Are specific protein-protein, protein-RNA, and RNA-RNA interactions strong enough to lead to the magic-ratio effect? The onset of the effect in our simulations occurs around $U_0 = 9k_BT$ (*Figure 3A*), which corresponds to a sticker-sticker dissociation constant $K_d = 0.4\,\text{mM}$. This value is consistent with the onset $K_d$ of 1–2.5 mM estimated from 3D lattice simulations with one polymer and one rigid component (*Xu et al., 2020*). For comparison, the measured $K_d$ for a SUMO protein domain with a SIM peptide is 0.01 mM (*Banani et al., 2016*) and the $K_d$ for an SH3 domain and a PRM peptide is 0.35 mM (*Li et al., 2012*). Thus for systems as strongly interacting as SUMO-SIM or SH3-PRM, the magic-ratio effect in principle should manifest in their phase diagrams. However, the magic-ratio effect has not been observed in these systems (*Li et al., 2012*; *Banani et al., 2016*), possibly due to size and linker length mismatch between the two associating polymers. Furthermore, real biological systems are more complex than our simple model. For example, there can be multiple-to-one binding, multiple

components, and the spacers/linkers can also play nontrivial roles (*Banjade et al., 2015*; *Harmon et al., 2017*). Currently, the in vivo relevance of the effects explored in this work remains an open question. Magic-ratio effects could also manifest in other experimental systems, such as non-biological polymers, DNA origami (*Hu and Niemeyer, 2019*), or patchy colloid systems (*Bianchi et al., 2011*). As an inverse problem, the magic-ratio effect could be exploited to determine the relative valence of associating biomolecules by measuring their phase diagram.

The magic-ratio effect allows for novel mechanisms of regulation. Chemical modifications, such as phosphorylation or SUMOylation, which change the effective valence of one component into or out of a magic-ratio condition could shift the phase boundary as a means of condensate regulation. Cells may also have evolved to avoid magic ratios so as to better promote condensate formation. For example, EPYC1 has valence five and Rubisco has valence eight, and the geometry of binding sites on Rubisco and the length of linkers in EPYC1 are such that they disfavor fully-bonded Rubisco-EPYC1 dimers even at equal polymer stoichiometry, which suppresses the magic-ratio effect (*He et al., 2020*). However, active removal of a terminal EPYC1 binding site, for example by phosphorylation (*Turkina et al., 2006*), would dramatically change the valence ratio to 1:2, which would then favor stable trimer formation, as previously suggested (*Freeman Rosenzweig et al., 2017*). We hope that our work will stimulate exploration of magic-ratio effects in both natural and synthetic multivalent, multicomponent systems.

The simulations and theory presented here are aimed at providing conceptual insights into the phase separation of associating polymers that form one-to-one specific bonds. Quantitative descriptions of related real systems will likely require additional features, such as details of molecular shape and flexibility, linker lengths, as well as range and type of interactions. For example, while the magic-ratio effect is robust with respect to the strength of nonspecific interactions and linker length, these variables do strongly influence phase boundaries. The dilute-phase concentrations in our simulations are ~mM, while the reported values for biological systems are typically tens of μM or less. The discrepancy is likely due to different strengths of nonspecific attraction, different length scales of steric replusion between stickers, and/or different lengths and flexibilities of the linkers (*Bhandari et al., 2021*). Indeed, increasing the nonspecific attraction in our simulations by a small amount $0.07k_{\mathrm{B}}T$ leads to a 50% reduction in the dilute-phase concentration (*Appendix 1—figure 4A*). Reducing the steric repulsion between beads of the same type has a similar effect (*Appendix 1—figure 5A*). More significantly, increasing the mean linker length from 4.7 nm to 5.9 nm leads to a more than 10-fold reduction in the dilute-phase concentration (*Appendix 1—figure 6A*). On the other hand, the dense-phase concentration strongly depends on the steric repulsion — increasing the sticker size from 2.5 to 2.9 nm decreases the dense phase concentration by a factor of 2 (*Appendix 1—figure 5B*). This is consistent with results from previous studies on the role of linkers: a self-avoiding random coil linker which occupies a large volume can substantially lower the dense-phase concentration and even prevent phase separation (*Harmon et al., 2017*). Future work will explore the interplay between specific and nonspecific interactions, and other molecular properties, and their roles in determining the physical properties of droplets, such as surface tension, viscosity, and rate of exchange between phases.

## Acknowledgements

We thank Guanhua He, Martin Jonikas, and Daniel Lee for insightful suggestions and comments. This work was supported in part by the National Science Foundation, through the Center for the Physics of Biological Function (PHY-1734030), and NSF grant PHY-1521553.

## Additional information

### Funding

| Funder | Grant reference number | Author |
| --- | --- | --- |
| National Science Foundation | PHY 1734030 | Yaojun Zhang<br>Bin Xu<br>Benjamin G Weiner<br>Yigal Meir<br>Ned S Wingreen |

| National Science Foundation | PHY 1521553 | Benjamin G Weiner |
| | | Ned S Wingreen |

The funders had no role in study design, data collection and interpretation, or the decision to submit the work for publication.

## Author contributions

Yaojun Zhang, Conceptualization, Data curation, Formal analysis, Writing - original draft, Writing - review and editing; Bin Xu, Yigal Meir, Conceptualization, Writing - review and editing; Benjamin G Weiner, Writing - review and editing; Ned S Wingreen, Conceptualization, Supervision, Funding acquisition, Writing - original draft, Writing - review and editing

## Author ORCIDs

Yaojun Zhang (iD) https://orcid.org/0000-0003-4587-6834
Benjamin G Weiner (iD) http://orcid.org/0000-0002-1995-8660
Ned S Wingreen (iD) https://orcid.org/0000-0001-7384-2821

## Decision letter and Author response

Decision letter https://doi.org/10.7554/eLife.62403.sa1
Author response https://doi.org/10.7554/eLife.62403.sa2

## Additional files

### Supplementary files

- Transparent reporting form

### Data availability

Codes for generating data in this manuscript can be found at https://github.com/yaojunz/matlab-lammps-phasediagram/tree/codes (copy archived at https://archive.softwareheritage.org/swh:1:rev:eff8367b00e1d12c17542bb9d03d85960a3e53e8/).

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

# Appendix 1

## Modeling two-component multivalent associative polymers

We perform coarse-grained molecular-dynamics simulations using LAMMPS (*Plimpton, 1995*) to simulate two-component multivalent associative polymers. Individual polymers are modeled as linear chains of spherical particles connected by harmonic bonds (*Appendix 1—figure 1A*, type A polymer in blue and type B polymer in yellow). Bonds are modeled using a harmonic potential (*Appendix 1—figure 1C*, *left*)

$$U_{\mathrm{b}}(r) = k(r - r_{\mathrm{b}})^2, \tag{16}$$

where $r_{\mathrm{b}} = 4.5$ nm is the mean bond length, $k = 20 k_{\mathrm{B}} T / r_{\mathrm{b}}^2$ is the bond stiffness, $k_{\mathrm{B}}$ is the Boltzmann constant, and T = 300 k is room temperature.

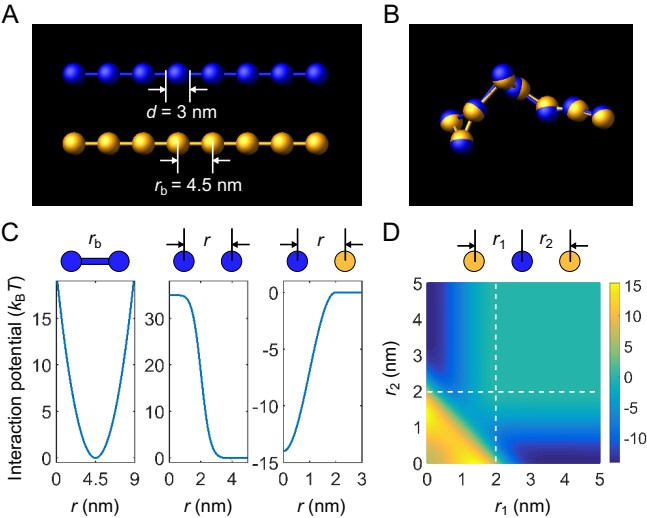

**Appendix 1—figure 1.** Coarse-grained molecular-dynamics simulations of two-component multivalent associative polymers. (**A**) Polymers are modeled as linear chains of spherical particles connected by harmonic bonds. Depicted are $A_8$ (blue) and $B_8$ (yellow). (**B**) Snapshot of a dimer of $A_8$ and $B_8$ formed in the strong-binding regime. (**C**) Neighboring stickers in a polymer are connected through a harmonic potential (*left*). Stickers of the same type interact pairwisely through a repulsive potential (*middle*). Stickers of different types interact pairwisely through an attractive potential (*right*). (**D**) Interaction energy between three particles (one A and two B stickers) as a function of their separation distances. Simultaneous binding of two B stickers to one A sticker is energetically highly disfavored (lower left region) compared to one-to-one binding (dark blue regions).

Stickers of the same type interact through a softened, truncated Lennard-Jones potential (*Appendix 1—figure 1C*, *middle*) [11]See LAMMPS manual at https://lammps.sandia.gov/doc/Manual.html for details about this potential.

$$U_{\mathrm{r}}(r) = 4\epsilon\lambda \left\{ \left[ (1-\lambda)^2 + \left(\frac{r}{\sigma}\right)^6 \right]^{-2} - \left[ (1-\lambda)^2 + \left(\frac{r}{\sigma}\right)^6 \right]^{-1} \right\}, \qquad r < r_{\mathrm{c}}, \tag{17}$$

where $\epsilon = 0.15 k_{\mathrm{B}} T$, $\lambda = 0.68$, $\sigma = 3.5\,\mathrm{nm}$, and $r_{\mathrm{c}} = 5\,\mathrm{nm}$. These parameters effectively lead to a sticker of diameter $d \simeq 3\,\mathrm{nm}$ and a weak attractive tail of depth $0.06 k_{\mathrm{B}} T$. The weak attractive tail is employed solely to promote a more compact dense condensate.

Stickers of different types interact through an attractive potential (*Appendix 1—figure 1C*, *right*)

$$U_{\mathrm{a}}(r) = -\frac{1}{2} U_0 \left( 1 + \cos\frac{\pi r}{r_0} \right), \qquad r < r_0 \tag{18}$$

where $U_0 = 14 k_{\mathrm{B}} T$ is used in all simulations in the main text, except as indicated in *Figure 3*. The

attraction cut-off distance is $r_0 = 2\,\text{nm}$. Note that due to the strong repulsion between stickers of the same type, simultaneous binding of two stickers of one type to a sticker of the other type is energetically highly disfavored (*Appendix 1—figure 1D*). This ensures one-to-one binding of stickers of different types (*Appendix 1—figure 1B*). Across our simulations, on average the fraction of stickers that have more than one partner is less than 0.001%.

## Phase equilibration and data recording

Each system consists of $n_1$ and $n_2$ polymers of types A and B, respectively. The number of polymers are determined by their valences/lengths ($L_1$ and $L_2$) and global A:B sticker stoichiometry ($s$) through

$$n_1 = \text{round}\left(\frac{Ns}{L_1(1+s)}\right),$$ (19)

$$n_2 = \text{round}\left(\frac{N}{L_2(1+s)}\right).$$ (20)

The round function is used as we can only simulate an integer number of polymers. For all simulations except those in *Figure 5* in the main text, we use $N = 2500$, so the total number of stickers is around 2500.

Simulations are equilibrated using a Langevin thermostat in the *NVT* ensemble at T = 300 K in a box of size 250 nm×50 nm×50 nm with periodic boundary conditions, that is the system evolves according to *Langevin, 1908*:

$$m\frac{d^2\vec{r}_i}{dt^2} = -\gamma\frac{d\vec{r}_i}{dt} - \nabla_{\vec{r}_i}U(\vec{r}_1,...,\vec{r}_N) + \vec{f}.$$ (21)

where $\vec{r}_i$ is the coordinate of particle $i$, $m$ is its mass, $\gamma$ is the friction coefficient, $\vec{f}$ is random thermal noise, and the energy $U(\vec{r}_1,...,\vec{r}_N)$ contains all interactions between particles, including harmonic bonds, nonspecific, and specific interactions (*Equations (16–18)*).

To promote phase equilibrium and ensure that only a single dense condensate is formed, we first initialize the simulation by confining polymers in the region $-50\,\text{nm}<x<50\,\text{nm}$. The attractive interaction between A and B stickers (*Equations (18)*) is gradually switched on from $U_0 = 0$ to 14 over $10^8$ time steps. The Langevin thermostat is applied using a damping factor $\tau = m/\gamma = 125\,\text{ns}$, step size $dt = 2.5\,\text{ns}$, and mass of particle m=3534.3 ag during this time period. These parameters give the particle the right diffusion coefficient $D = k_\text{B}T/(3\pi\eta d)$ for times longer than $\tau$, where $\eta$ is the water viscosity 0.001 kg/m/s and $d$ the diameter of the particle. This annealing procedure leads to the formation of a dense phase close to its equilibrated concentration. The confinement is then removed, and the system is equilibrated for $10^8$ more time steps to allow the formation of dilute phase and relaxation of the dense phase. After these procedures, we switch to smaller τ=10 ns, dt=0.5 ns, and m=282.7 ag for data recording ($D$ remains the same). The system is relaxed for another $10^8$ steps. The relaxation time of the system depends on the sticker-sticker bond lifetime; to ensure that the dilute and dense phases are in equilibrium, the above choice of relaxation time before recording corresponds to ~2000 bond lifetimes. We then recorded the positions of all particles every $10^6$ steps for 400 recordings. For each choice of valence and stoichiometry, we performed 10 simulation replicates with different random seeds. We also checked whether there are systematic deviations between the first and second halves of the recorded simulations, and found consistent results between the two halves.

To test the effect of finite size on the phase boundaries, simulations in *Figure 5* in the main text are performed with $N = 5000$ and a box of size 315 nm×63 nm×63 nm, that is both total number of stickers and box volume are doubled while the total global sticker concentration remains the same (6.64 mM). Procedures for equilibration and data recording are the same (including the initial confinement region) except systems are relaxed for $5 \times 10^8$ steps at $dt = 0.5\,\text{ns}$ before recording, as the larger system requires a longer relaxation time.

## Determining the phase boundaries

To determine the phase boundaries, we need to obtain the concentrations in the dilute and dense phases. The data we recorded for each system contains 4000 snapshots of polymer configurations (10 replicates and 400 time points each). To measure concentrations, we first group polymers into clusters in each snapshot. Connected stickers are grouped into one cluster: two stickers of the same type are connected if they are neighbors in the same polymer, and two stickers of different types are connected if they are within the attraction distance $r_0 = 2$ nm. In most of our simulations, in each snapshot, we observe one large cluster which contains most of polymers, and a few to tens of very small clusters (*Appendix 1—figure 2*). There is a clear gap between the sizes of these large and small clusters. We define the large cluster as the dense phase, and the smaller clusters as constituents of the dilute phase. In cases where the separation between the dense and dilute phases is unclear, we discard the data set. *Appendix 1—figure 2* shows the size distribution of clusters pooled over all snapshots. *Appendix 1—table 1* lists all the dilute-phase components and their total sticker percentages in the $A_{14}:B_{14}$ system at global sticker stoichiometry 1.21.

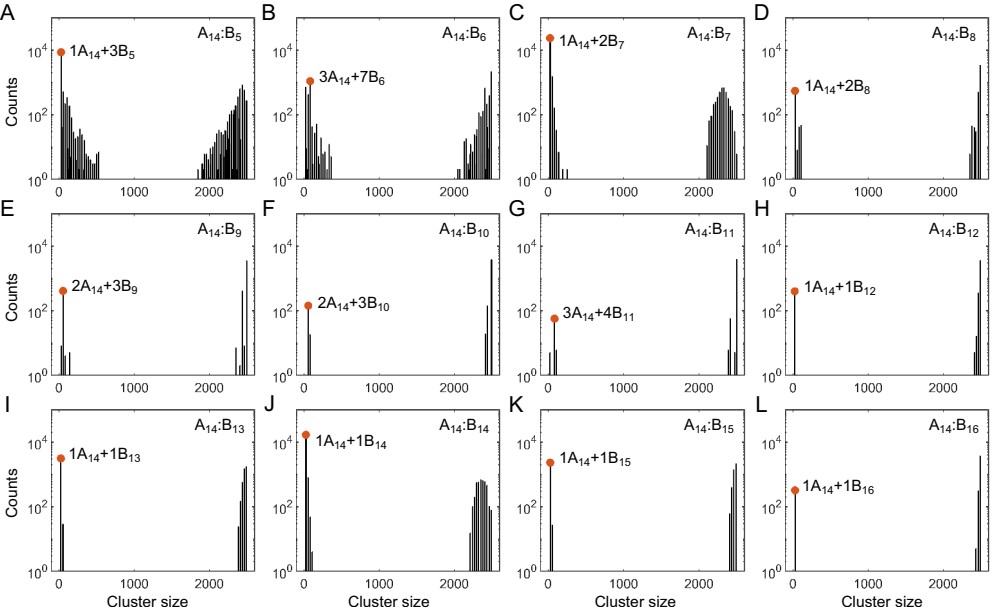

**Appendix 1—figure 2.** Histograms of cluster size in stickers in (**A–L**) $A_{14}:B_{5-16}$ systems at equal global sticker stoichiometry. Parameters: specific binding strength $U_0 = 14 k_B T$ and total global sticker concentration 6.64 mM. Red dots indicate the dominant oligomer in the dilute phase.

**Appendix 1—table 1.** List of all dilute-phase components and sticker percentages for $A_{14}:B_{14}$ system at global sticker stoichiometry 1.21.

| Comp | $1A_{14}$ | $1A_{14}+1B_{14}$ | $2A_{14}+1B_{14}$ | $2A_{14}+2B_{14}$ | $3A_{14}+2B_{14}$ | $4A_{14}+3B_{14}$ | $5A_{14}+3B_{14}$ |
|---|---|---|---|---|---|---|---|
| frac | 0.10% | 37.0% | 7.67% | 0.38% | 19.1% | 16.6% | 0.38% |
| comp | $5A_{14}+4B_{14}$ | $6A_{14}+4B_{14}$ | $6A_{14}+5B_{14}$ | $7A_{14}+5B_{14}$ | $8A_{14}+6B_{14}$ | $10A_{14}+7B_{14}$ | $17A_{14}+13B_{14}$ |
| frac | 10.8% | 0.85% | 0.42% | 0.80% | 3.45% | 0.48% | 1.99% |

To find the dilute- and dense-phase concentrations, we calculate the center of mass of the dense cluster for each snapshot, and recenter the simulation box to this center of mass. We then compute the sticker concentration histogram along the $x$ axis with a bin size 1/50 of box length. The resulting concentration profile has high values in the middle corresponding to the dense-phase concentration, and low values on the two sides corresponding to the dilute-phase concentration. The dilute- and dense-phase concentrations are calculated by averaging the concentration profile over the regions ($x \leq -100$ nm or $x \geq 100$ nm) and ($-10$ nm $\leq x \leq 10$ nm), respectively.

## Testing the effect of finite simulation size on the phase boundaries

To test the effect of finite size on the phase boundaries, we compare the dilute- and dense-phase concentrations of systems with $N = 2500$ to systems with $N = 5000$ (*Appendix 1—table 2*). The systems being compared have the same valence, stoichiometry, and total sticker concentration. Simulations with different total numbers of particles show consistent results, suggesting the effect of finite size is minor.

**Appendix 1—table 2.** Comparison of dilute- and dense-phase concentrations for systems with different total numbers of particles.

| Valence (Stoichiometry) | $A_{14}:B_{14}$ (14:14) | $A_{14}:B_{13}$ (14:14) | $A_{14}:B_{13}$ (14:13) |
|---|---|---|---|
| $c_{dil}$ (mM) for $N = 2500$ | $0.50 \pm 0.06$ | $0.08 \pm 0.02$ | $0.39 \pm 0.02$ |
| $c_{dil}$ (mM) for $N = 5000$ | $0.45 \pm 0.03$ | $0.12 \pm 0.02$ | $0.34 \pm 0.03$ |
| $c_{dil}$ (mM) for $N = 2500$ | $28.0 \pm 0.2$ | $27.76 \pm 0.09$ | $25.80 \pm 0.08$ |
| $c_{den}$ (mM) for $N = 5000$ | $27.93 \pm 0.05$ | $27.82 \pm 0.06$ | $25.82 \pm 0.06$ |

## Determining the percolation threshold

Phase transitions in associative polymeric systems can be thought as phase separation aided percolation, that is, the dense phase of an associative polymer system is a percolating network *Semenov and Rubinstein, 1998*; *Harmon et al., 2017*; *Choi et al., 2020a*. What would be the percolation threshold in our associative polymer system if the density remained homogeneous? To answer this question, we determined the percolation threshold of an $A_8:B_8$ system at a weak binding strength $U_0 = 8k_BT$, which avoids phase separation. Briefly, for a given sticker concentration $(c_A, c_B)$, we perform one simulation at $U_0 = 8k_BT$ and analyze the size of clusters for all snapshots recorded after the system equilibrates. We judge whether polymers are in a sol- or gel-state based on the following gelation criterions: First, for each snapshot if the largest cluster contains more than 70% of the stickers and the second largest cluster contains less than 10% of the stickers in the system, we label this snapshot as having a percolating cluster. Second, for a given system if more than 50% of snapshots have a percolating cluster, we label this $(c_A, c_B)$ point as a 'gel' state. The systems do not meet the gelation criterions are labeled as a 'sol' state (*Appendix 1—figure 3*). It is clear that the dilute/dense phases of the $A_8:B_8$ system formed at a strong binding strength $U_0 = 14k_BT$ are in a sol-/gel-state. By inspection, the same clear dichotomy applies to the other systems we simulated.

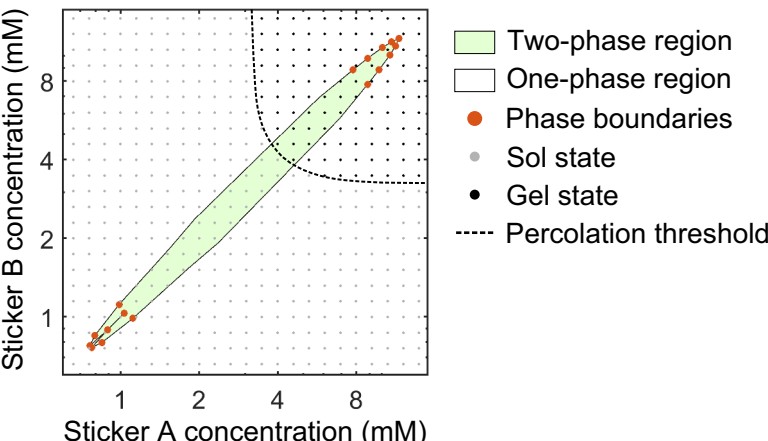

**Appendix 1—figure 3.** Phase diagram and percolation threshold of $A_8:B_8$ system. Phase boundaries (red dots) are measured from MD simulations at $U_0 = 14k_BT$. The complete two-phase region (green) and one-phase region (white) are extrapolated based on the phase boundaries from simulations. Sol- (gray dots) and gel- (black dots) states from simulations at $U_0 = 8k_BT$ where the system remains homogeneous are identified based on the gelation criterions. Percolation threshold (dashed line) is interpolated from the labeled sol- and gel-states.

## Measuring the dissociation constants

The phase boundaries of associative polymers from theory are very sensitive to model parameters. Here, we extract the dimer and sticker dissociation constants $K_d$ and $K_b$ from simulations. These values are then utilized in the dimer-gel theory to obtain the free-energy density landscape and predict the phase boundaries. Our simulations are performed in the strong binding regime. At our chosen binding strength of $U_0 = 14k_BT$, the long lifetime of each bond means that dimers of valence $\geq 4$ never dissociate in our simulations. This prevents us from directly extracting the dimer dissociation constants for long polymers. We therefore use a reweighting method *Frenkel and Smit, 2001* to obtain the dissociation constant for these dimers.

Briefly, we perform simulations with 1 polymer of type A and 1 polymer of type B of valences $L_1$ and $L_2$ in a cubic box of side $20L_1$ nm with periodic boundary conditions. The systems are equilibrated using a Langevin thermostat in the *NVT* ensemble, all the parameters are the same as the ones used for recording the data, except here we use $U_0 = 7k_BT$ which is half of the original value, in order to allow dimers to dissociate.

Theoretically, the dissociation constant of a dimer is defined as:

$$K_d^{-1} = V \frac{\iint_{U_{AB}<0} e^{-\beta U_A(\vec{r}_1^A,\dots,\vec{r}_{L_1}^A)} e^{-\beta U_B(\vec{r}_1^B,\dots,\vec{r}_{L_2}^B)} e^{-\beta U_{AB}(\vec{r}_1^A,\dots,\vec{r}_{L_1}^A,\vec{r}_1^B,\dots,\vec{r}_{L_2}^B)} d\vec{r}_1^A \dots d\vec{r}_{L_1}^A d\vec{r}_1^B \dots, d\vec{r}_{L_2}^B}{\iint e^{-\beta U_A(\vec{r}_1^A,\dots,\vec{r}_{L_1}^A)} e^{-\beta U_B(\vec{r}_1^B,\dots,\vec{r}_{L_2}^B)} d\vec{r}_1^A \dots d\vec{r}_{L_1}^A d\vec{r}_1^B \dots, d\vec{r}_{L_2}^B}, \tag{22}$$

where $\beta = 1/k_BT$, and $(\vec{r}_1^A,\dots,\vec{r}_{L_1}^A)$ and $(\vec{r}_1^B,\dots,\vec{r}_{L_2}^B)$ are the coordinates of stickers in polymers A and B. $U_A(U_B)$ contain all interactions within the polymer A(B), including bond potentials $U_b$ (*Equations (16)*) and nonspecific interactions $U_r$ (*Equations (17)*). $U_{AB}$ contains all the specific interactions between polymers A and B, that is the sum of $U_a$s (*Equations (18)*). Integration is over the entire volume $V$. In the numerator, the integration is further confined to the region where $U_{AB}<0$. Note that for a dissociated dimer $U_{AB} = 0$.

To link the simulation with the definition of $K_d$, we define three variables in the simulation:

$$w_1 = e^{\beta E_{AB}}, w_2 = 1, w_3 = e^{-\beta E_{AB}}, \qquad \text{if } E_{AB}<0;$$
$$w_1 = w_2 = w_3 = 0, \qquad \text{if } E_{AB} = 0, \tag{23}$$

where $E_{AB} = U_{AB}(\vec{r}_1^A,\dots,\vec{r}_{L_1}^A,\vec{r}_1^B,\dots,\vec{r}_{L_2}^B)$. We then have,

$$K_d^{-1}(U_0 = 0k_BT) = C \iint w_1 e^{-\beta U_A} e^{-\beta U_B} e^{-\beta U_{AB}} d\vec{r}_1^A \dots d\vec{r}_{L_1}^A d\vec{r}_1^B \dots, d\vec{r}_{L_2}^B = \tilde{C}\langle w_1 \rangle, \tag{24}$$

$$K_d^{-1}(U_0 = 7k_BT) = C \iint w_2 e^{-\beta U_A} e^{-\beta U_B} e^{-\beta U_{AB}} d\vec{r}_1^A \dots d\vec{r}_{L_1}^A d\vec{r}_1^B \dots, d\vec{r}_{L_2}^B = \tilde{C}\langle w_2 \rangle, \tag{25}$$

$$K_d^{-1}(U_0 = 14k_BT) = C \iint w_3 e^{-\beta U_A} e^{-\beta U_B} e^{-\beta U_{AB}} d\vec{r}_1^A \dots d\vec{r}_{L_1}^A d\vec{r}_1^B \dots, d\vec{r}_{L_2}^B = \tilde{C}\langle w_3 \rangle, \tag{26}$$

where $\langle w_1 \rangle$, $\langle w_2 \rangle$, and $\langle w_3 \rangle$ are the mean values of $w_1$, $w_2$, and $w_3$ obtained by averaging over a simulation with $U_0 = 7k_BT$. $C$ and $\tilde{C}$ are constants and are the same in all three equations. Therefore,

$$K_d(U_0 = 0k_BT)\langle w_1 \rangle = K_d(U_0 = 7k_BT)\langle w_2 \rangle = K_d(U_0 = 14k_BT)\langle w_3 \rangle. \tag{27}$$

On the other hand, it can be shown that the binding probability $\langle w_2 \rangle$ for $U_0 = 7k_BT$ is

$$\langle w_2 \rangle = \frac{\iint w_2 e^{-\beta U_A} e^{-\beta U_B} e^{-\beta U_{AB}} d\vec{r}_1^A \dots d\vec{r}_{L_1}^A d\vec{r}_1^B \dots, d\vec{r}_{L_2}^B}{\iint e^{-\beta U_A} e^{-\beta U_B} e^{-\beta U_{AB}} d\vec{r}_1^A \dots d\vec{r}_{L_1}^A d\vec{r}_1^B \dots, d\vec{r}_{L_2}^B} = \frac{1}{1 + K_d(U_0 = 7)(V - K_d^{-1}(U_0 = 0))}. \tag{28}$$

Combining *Equations (27) and (28)*, we have

$$K_d(U_0 = 0k_BT) = \frac{1 + \langle w_1 \rangle - \langle w_2 \rangle}{\langle w_1 \rangle V}, \tag{29}$$

$$K_\mathrm{d}(U_0 = 7k_\mathrm{B}T) = \frac{1 + \langle w_1 \rangle - \langle w_2 \rangle}{\langle w_2 \rangle V}, \tag{30}$$

$$K_\mathrm{d}(U_0 = 14k_\mathrm{B}T) = \frac{1 + \langle w_1 \rangle - \langle w_2 \rangle}{\langle w_3 \rangle V}. \tag{31}$$

*Appendix 1—table 3* shows a list of dissociation constants from simulations. The reweighting method provides very accurate sticker-sticker and dimer-dimer dissociation constants, as confirmed by comparing with theory and direct simulations for stickers and polymers of length $L = 2$. The dissociation constants obtained by the reweighting method are used in the dimer-gel theory to predict the phase boundaries (Appendix 2).

**Appendix 1—table 3.** List of dissociation constants from simulations.

|  | $A_1{:}B_1$ (t) | $A_1{:}B_1$ (d) | $A_1{:}B_1$ (r) | $A_2{:}B_2$ (d) | $A_2{:}B_2$ (r) |
|---|---|---|---|---|---|
| $K_\mathrm{d}(U_0 = 0k_\mathrm{B}T)$ (mM) | 49.6 | * | 48.9 | * | 12.4 |
| $K_\mathrm{d}(U_0 = 7k_\mathrm{B}T)$ (mM) | 1.88 | * | 1.90 | * | 0.31 |
| $K_\mathrm{d}(U_0 = 14k_\mathrm{B}T)$ (mM) | 5.7e-3 | 5.9e-3 | 5.8e-3 | 7.7e-6 | 7.1e-6 |
|  | $A_4{:}B_3$ (r) | $A_4{:}B_4$ (r) | $A_6{:}B_5$ (r) | $A_6{:}B_6$ (r) |  |
| $K_\mathrm{d}(U_0 = 0k_\mathrm{B}T)$ (mM) | 4.32 | 3.29 | 1.81 | 1.52 |  |
| $K_\mathrm{d}(U_0 = 7k_\mathrm{B}T)$ (mM) | 4.5e-2 | 2.2e-2 | 4.3e-3 | 2.2e-3 |  |
| $K_\mathrm{d}(U_0 = 14k_\mathrm{B}T)$ (mM) | 3.77e-9 | 1.38e-11 | 6.95e-15 | 3.00e-17 |  |
|  | $A_8{:}B_6$ (r) | $A_8{:}B_7$ (r) | $A_8{:}B_8$ (r) | $A_8{:}B_9$ (r) | $A_8{:}B_{10}$ (r) |
| $K_\mathrm{d}(U_0 = 0k_\mathrm{B}T)$ (mM) | 1.15 | 1.01 | 0.90 | 0.79 | 0.73 |
| $K_\mathrm{d}(U_0 = 7k_\mathrm{B}T)$ (mM) | 9.0e-4 | 4.5e-4 | 2.5e-4 | 1.5e-4 | 1.0e-4 |
| $K_\mathrm{d}(U_0 = 14k_\mathrm{B}T)$ (mM) | 4.22e-18 | 1.26e-20 | 7.07e-23 | 1.77e-23 | 7.59e-24 |
|  | $A_{14}{:}B_{12}$ (r) | $A_{14}{:}B_{13}$ (r) | $A_{14}{:}B_{14}$ (r) | $A_{14}{:}B_{15}$ (r) | $A_{14}{:}B_{16}$ (r) |
| $K_\mathrm{d}(U_0 = 0k_\mathrm{B}T)$ (mM) | 0.37 | 0.32 | 0.30 | 0.32 | 0.27 |
| $K_\mathrm{d}(U_0 = 7k_\mathrm{B}T)$ (mM) | 1.4e-6 | 6.7e-7 | 3.8e-7 | 2.6e-7 | 1.5e-7 |
| $K_\mathrm{d}(U_0 = 14k_\mathrm{B}T)$ (mM) | 2.74e-35 | 1.08e-37 | 8.85e-40 | 1.80e-40 | 5.16e-41 |

*footnotetext: (t) theoretical value, (d) direct simulation, and (r) reweighting method.

## Effects of nonspecific interactions and linker length

In the main text, we focused on the effects of binding strength, sticker stoichiometry, and polymer valences on the phase boundaries of two-component systems. Here, we explore the effects of nonspecific interactions and linker length. Increasing the nonspecific attraction between stickers of same type leads to decreased/increased dilute-/dense-phase concentrations (*Appendix 1—figure 4*). The additional attractive interaction is modeled by a cosine-squared potential

$$U(r) = \begin{cases} -\epsilon, & r < \sigma, \\ -\epsilon \cos\left(\frac{\pi(r-\sigma)}{2(r_c - \sigma)}\right)^2, & \sigma \le r < r_c, \\ 0, & r \ge r_c. \end{cases} \tag{32}$$

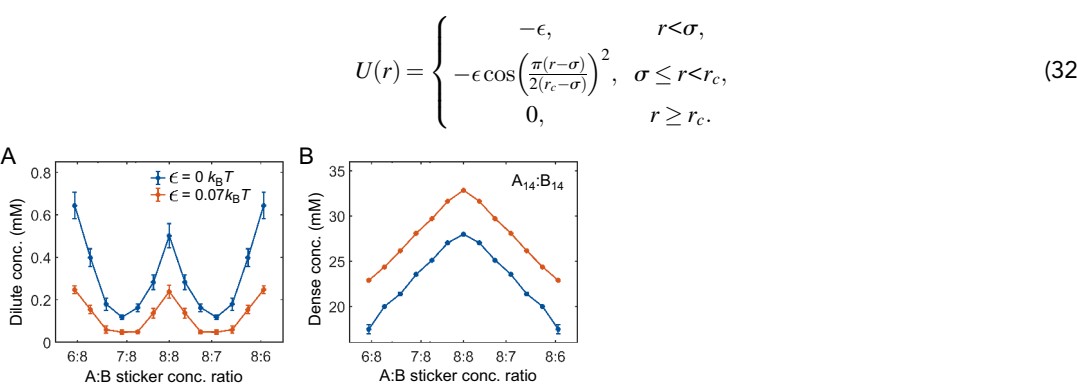

**Appendix 1—figure 4.** Strength of nonspecific attractive interactions strongly influences simulated

*Appendix 1—figure 4 continued on next page*

*Appendix 1—figure 4 continued*

dilute-phase boundary. Total sticker concentrations in (**A**) dilute and (**B**) dense phases for $A_{14}:B_{14}$ system at different global sticker stoichiometries with the additional attractive interaction strength $\epsilon = 0k_BT$ (blue) and $\epsilon = 0.07k_BT$ (red) ($\sigma = 3.5\,\text{nm}$ and $r_c = 5\,\text{nm}$ in *Equation (32)*). Parameters: specific interaction strength $U_0 = 14k_BT$, total global sticker concentration 6.64 mM.

The cosine-squared potential is applied together with the softened, truncated Lennard-Jones potential (*Equation (17)*). Similarly, reducing the range of repulsion between stickers of the same type leads to decreased/increased dilute-/dense-phase concentrations (*Appendix 1—figure 5*). Here, the repulsive interaction potential between same type of stickers is replaced by the standard repulsive Lennard-Jones potential

$$U_r(r) = 4\epsilon\left[\left(\frac{\sigma}{r}\right)^{12} - \left(\frac{\sigma}{r}\right)^6 + \frac{1}{4}\right], \qquad r < 2^{1/6}\sigma. \qquad (33)$$

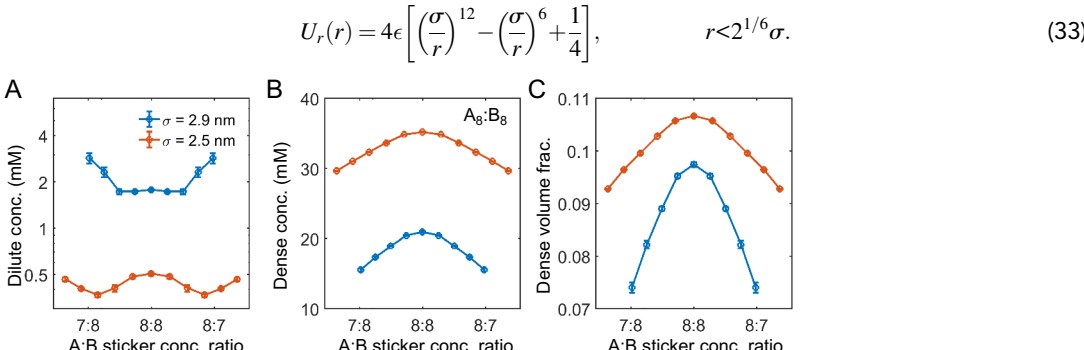

**Appendix 1—figure 5.** Range of nonspecific repulsive interactions strongly influences phase boundaries in simulations. Sticker concentrations in (**A**) dilute and (**B**) dense phases and (**C**) volume fraction of polymers in dense phase for $A_8:B_8$ system at different global sticker stoichiometries with standard repulsive Lennard-Jones potential (*Equation (33)*) σ = 2.9 nm (blue) and σ = 2.5 nm (red). Parameters: repulsive interaction strength $\epsilon = 1k_BT$, specific interaction strength $U_0 = 12k_BT$ with cut-off distance $r_0$ = 1.5 nm, and total global sticker concentration is 6.64 mM. Note that the overlapping volume between two bound stickers is counted once in the volume fraction calculation, that is, two perfectly overlapping stickers only occupy a volume of one sticker.

The phase boundaries are also sensitive to the linker length. Here, we model the repulsive interactions between same type stickers by *Equation (33)*, the bond between stickers by an expanded FENE potential

$$U_b(r) = -\frac{1}{2}KR_0^2\ln\left[1 - \left(\frac{r-\sigma}{R_0}\right)^2\right], \qquad r < \sigma + R_0, \qquad (34)$$

and attraction between different types of stickers by *Equation (18)*. Increasing of mean bond length from 4.7 to 5.9 nm leads to a decrease of the dilute-phase concentration by more than a factor of 10 (*Appendix 1—figure 6*).

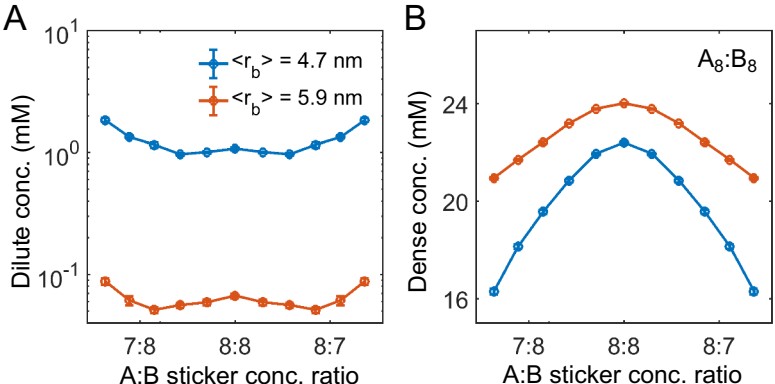

*Appendix 1—figure 6 continued on next page*

*Appendix 1—figure 6 continued*

**Appendix 1—figure 6.** Linker length strongly influences phase boundaries in simulations. Sticker concentrations in (**A**) dilute and (**B**) dense phases for $A_8$:$B_8$ system at different global sticker stoichiometries with linkers modeled by a FENE potential (*Equation (34)*) with $K = 0.3k_\mathrm{B}T$ and $R_0$ = 7 nm (mean linker length 4.7 nm, blue) and $K = 0.15k_\mathrm{B}T$ and $R_0$ = 14 nm (mean linker length 5.9 nm, red). Parameters: repulsive interaction strength $\epsilon = 1k_\mathrm{B}T$ and length scale $\sigma$ = 3 nm, specific interaction strength $U_0 = 12k_\mathrm{B}T$ with cut-off distance $r_0$ = 1.5 nm, and total global sticker concentration 6.64 mM.

# Appendix 2

## Derivation of free-energy density for nonspecific interactions

The free-energy density due to nonspecific interactions can be written as a power expansion in the concentrations *Semenov and Rubinstein, 1998*; *De Gennes, 1979*,

$$\frac{F_{\mathrm{ns}}}{k_{\mathrm{B}}T} = \frac{1}{2}\sum_{ij} v_{ij}c_i c_j + \frac{1}{6}\sum_{ijk} w_{ijk}c_i c_j c_k, \tag{35}$$

where the sum is over all the species in the system, including free polymers/stickers, dimers and independent bonds, and $v_{ij}$ and $w_{ijk}$ are two- and three-body interaction parameters.

In the strong-binding regime where the magic-ratio effect is observed, bound stickers strongly overlap, so the size of a bound pair is almost that of a free sticker. For simplicity, we therefore assume the interactions between dimers to be the same as between free polymers of the same type (denoted as $v_{\mathrm{d}}$ and $w_{\mathrm{d}}$), and the interactions between independent bonds to be the same as between free stickers of the same type (denoted as $v_{\mathrm{b}}$ and $w_{\mathrm{b}}$). When independent bonds are preferred (i.e. when $\rho_{\mathrm{d}} \approx 0$), the free-energy density for nonspecific interactions is

$$\frac{F_{\mathrm{ns}}^{\mathrm{ind}}}{k_{\mathrm{B}}T} = \frac{v_{\mathrm{b}}}{2}\left[(c_1 - c_{\mathrm{b}})^2 + (c_2 - c_{\mathrm{b}})^2 + 2(c_1 - c_{\mathrm{b}})c_{\mathrm{b}} + 2(c_2 - c_{\mathrm{b}})c_{\mathrm{b}} + c_{\mathrm{b}}^2\right] +$$
$$\frac{w_{\mathrm{b}}}{6}\left[(c_1 - c_{\mathrm{b}})^3 + (c_2 - c_{\mathrm{b}})^3 + 3(c_1 - c_{\mathrm{b}})^2 c_{\mathrm{b}} + 3(c_1 - c_{\mathrm{b}})c_{\mathrm{b}}^2 + 3(c_2 - c_{\mathrm{b}})^2 c_{\mathrm{b}} + 3(c_2 - c_{\mathrm{b}})c_{\mathrm{b}}^2 + c_{\mathrm{b}}^3\right], \tag{36}$$

where $c_1$, $c_2$, and $c_{\mathrm{b}}$ are the total concentrations of polymer A, B, and independent bonds in sticker units. $c_1 - c_{\mathrm{b}}$ and $c_2 - c_{\mathrm{b}}$ are therefore the concentrations of free sticker A and B. Note that in our simulations, there is no nonspecific interaction between free polymers of different types. Therefore, all $v$ and $w$ terms involving different free species are 0. *Equation (36)* simplifies to

$$\frac{F_{\mathrm{ns}}^{\mathrm{ind}}}{k_{\mathrm{B}}T} = \frac{v_{\mathrm{b}}}{2}\left(c_1^2 + c_2^2 - c_{\mathrm{b}}^2\right) + \frac{w_{\mathrm{b}}}{6}\left(c_1^3 + c_2^3 - c_{\mathrm{b}}^3\right). \tag{37}$$

Similarly, when dimers are preferred (i.e. when $c_{\mathrm{b}} \approx 0$), the free-energy density for nonspecific interactions is

$$\frac{F_{\mathrm{ns}}^{\mathrm{dim}}}{k_{\mathrm{B}}T} = \frac{v_{\mathrm{d}}}{2}\left(\rho_1^2 + \rho_2^2 - \rho_{\mathrm{d}}^2\right) + \frac{w_{\mathrm{d}}}{6}\left(\rho_1^3 + \rho_2^3 - \rho_{\mathrm{d}}^3\right), \tag{38}$$

where $\rho_1$, $\rho_2$, and $\rho_{\mathrm{d}}$ are the total concentrations of polymer A, B, and dimers in polymeric units.

As nonspecific interactions are only important at high concentrations, we simply set $F_{\mathrm{ns}} = F_{\mathrm{ns}}^{\mathrm{ind}}$. Further, in the strong-binding regime, $c_{\mathrm{b}} \approx \min(c_1, c_2)$, so

$$\frac{F_{\mathrm{ns}}}{k_{\mathrm{B}}T} = \frac{v_{\mathrm{b}}}{2}\max(c_1, c_2)^2 + \frac{w_{\mathrm{b}}}{6}\max(c_1, c_2)^3. \tag{39}$$

## Determining model parameters

The developed dimer-gel theory has only four parameters for a given system $A_{L1}{:}B_{L2}$: the dissociation constants of dimers and independent bonds $K_{\mathrm{d}}$ and $K_{\mathrm{b}}$, and the nonspecific interaction parameters $v_{\mathrm{b}}$ and $w_{\mathrm{b}}$. The four parameters together determine the competitiveness of the dilute dimer-phase with the dense gel-phase.

We have extracted the values of $K_{\mathrm{d}}$ from simulations (*Appendix 1—table 3*). Physically, we expect $K_{\mathrm{b}}$ to be close to the sticker-sticker dissociation constant $5.7\,\mu\mathrm{M}$ (*Appendix 1—table 3*). It is not exactly the same because the bonds in the condensate are tethered by the backbones of the polymers. We expect the nonspecific interaction parameters to be approximately $v_{\mathrm{b}} = 2B_2$ and $w_{\mathrm{b}} = 3B_3$, where $B_2$ and $B_3$ are the second and third virial coefficients *Katsura, 1959*: $v_{\mathrm{b}} = 6.8 \times 10^{-2}\mathrm{mM}^{-1}$ and $w_{\mathrm{b}} = 2.2 \times 10^{-3}\mathrm{mM}^{-2}$ for hard spheres of diameter 3 nm (the size of a sticker in the simulations), respectively.

The predicted phase boundaries are sensitive to these parameters. We thus tune $K_b$, $v_b$, and $w_b$ around their estimated values to match the dilute- and dense-phase concentrations of the $A_8$:$B_8$ system from simulation. Specifically, we yield $K_b = 3.8 \times 10^{-3}$mM, $v_b = 9 \times 10^{-2}$mM$^{-1}$, and $w_b = 7 \times 10^{-3}$mM$^{-2}$. These parameters are used for $A_8$:$B_{6-10}$ systems. Further discussions on how model parameters affect phase boundaries can be found in later sections.

## Derivation of the transition concentration $c_s$

In the strong-binding regime, the free-energy density contributions from specific interactions in the dimer-dominated and independent bond-dominated limit are, respectively,

$$\frac{F_s^{dim}}{k_B T} = \rho_- \ln K_d + (\rho_+ - \rho_-) \ln \frac{\rho_+ - \rho_-}{e} - \rho_+ \ln \frac{\rho_+}{e}, \tag{40}$$

$$\frac{F_s^{ind}}{k_B T} = c_- \ln K_b + (c_+ - c_-) \ln \frac{c_+ - c_-}{e} - c_+ \ln \frac{c_+}{e}, \tag{41}$$

where $\rho_+ = \max(c_1/L_1, c_2/L_2)$, $\rho_- = \min(c_1/L_1, c_2/L_2)$, $c_+ = \max(c_1, c_2)$, and $c_- = \min(c_1, c_2)$. **Equations (40) and (41) are the same as Equations (14) and (15).**

At given $(c_1, c_2)$, whether the system will form dimers or independent bonds depends on their relative free energies. For equal valence polymers $L_1 = L_2 = L$, letting $c_+ = s c_-$, **Equations (40) and (41)** become

$$\frac{F_s^{dim}}{k_B T} = \rho_- \ln \frac{(s-1)^{s-1} e K_d}{s^s \rho_-}, \tag{42}$$

$$\frac{F_s^{ind}}{k_B T} = c_- \ln \frac{(s-1)^{s-1} e K_b}{s^s c_-}. \tag{43}$$

Comparing the two expressions, dimers are favored at low concentrations ($F_s^{dim} < F_s^{ind}$), and independent bonds are favored at high concentrations ($F_s^{ind} < F_s^{dim}$). The transition occurs when $F_s^{dim} = F_s^{ind}$, i.e. at the concentrations $c_-(s) = c_0 (s-1)^{s-1} s^{-s}$ and correspondingly $c_+(s) = c_0 s (s-1)^{s-1} s^{-s}$ where $c_0 = e(K_b^L/(K_d L))^{1/(L-1)}$. The boundary between dimer- and independent bond-dominated regions is described by $(c_+(s), c_-(s))$ and $(c_-(s), c_+(s))$, respectively, in the lower and upper halves of the $(c_1, c_2)$ plane (**Appendix 2–figure 1A**, white curve).

## Solving reaction *Equations (7) and (8)*

The high powers in **Equation (7)** and the small value of $K_d$ make it difficult to find numerical solutions of $c_d$ and $c_b$ accurately. To overcome this difficulty, we define a variable $\lambda = c_1 - c_{d1} - c_b$ when $\rho_1 \leq \rho_2$ and $\lambda = c_2 - c_{d2} - c_b$ when $\rho_1 > \rho_2$, and rewrite **Equations (7) and (8)** in terms of $\lambda$. Specifically,

$$c_b = \lambda (c_2 L_1 - c_1 L_2 + \lambda L_2) [K_b L_1 + \lambda (L_1 - L_2)]^{-1},$$
$$c_{d1} = c_1 - \lambda - c_b, \tag{44}$$
$$K_d L_2 (c_1 - c_b - \lambda)(1 + \lambda K_b^{-1})^{L_2 - 1}(\lambda + c_b)^{L_1 - 1} = K_b c_b \lambda^{L_1 - 1}$$

for $\rho_1 \leq \rho_2$, and

$$c_b = \lambda (c_1 L_2 - c_2 L_1 + \lambda L_1) [K_b L_2 + \lambda (L_2 - L_1)]^{-1},$$
$$c_{d2} = c_2 - \lambda - c_b, \tag{45}$$
$$K_d L_1 (c_2 - c_b - \lambda)(1 + \lambda K_b^{-1})^{L_1 - 1}(\lambda + c_b)^{L_2 - 1} = K_b c_b \lambda^{L_2 - 1}$$

for $\rho_1 > \rho_2$. We then solve **Equations (44) and (45)** using the MatLab function *vpasolve* with the constraints $0 < \lambda < c_1$ and $0 < \lambda < c_2$, respectively. *vpasolve* provides all solutions within the specified range. When multiple solutions coexist, we take the one with the lowest $F_s$. Numerical solutions of $\rho_d$ and $c_b$ for $A_8$:$B_8$ and $A_8$:$B_7$ systems are shown in **Appendix 2–figure 1**, where the fraction of stickers in

dimers is defined as $\rho_d/\min(\rho_1,\rho_2)$ and fraction of stickers in independent bonds is defined as $c_b/\min(c_1,c_2)$.

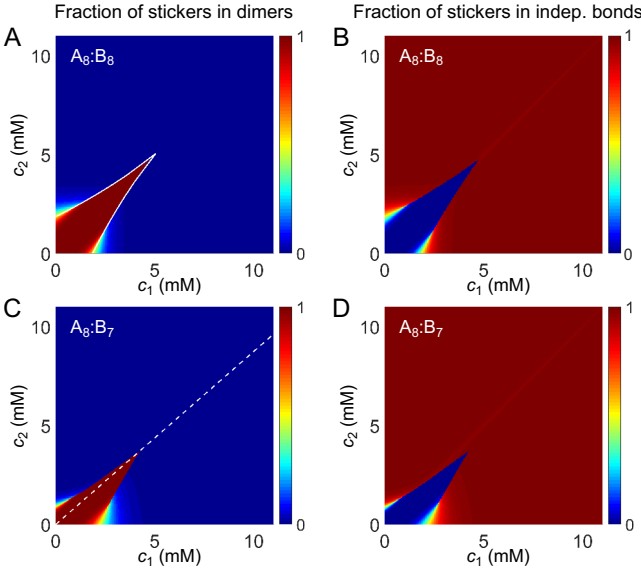

**Appendix 2—figure 1.** Fraction of stickers in dimers (**A**) and in independent bonds (**B**) for $A_8$:$B_8$ system. White curve in (**A**) is the transition boundary between dimer- and independent bonds-dominated regions predicted by $c_+(s)$ and $c_-(s)$. Fraction of stickers in dimers (**C**) and in independent bonds (**D**) for $A_8$:$B_7$ system. Dashed white line denotes equal polymer stoichiometry.

## Determining phase boundaries and tie lines

We obtain the free-energy landscape by substituting the numerical solutions of $\rho_d$ and $c_b$ into *Equation (13)* (*Appendix 2–figure 2*). We locate the phase boundaries by applying convex-hull analysis to this free-energy landscape using the MatLab function *convhull* (*Figure 6A and B*).

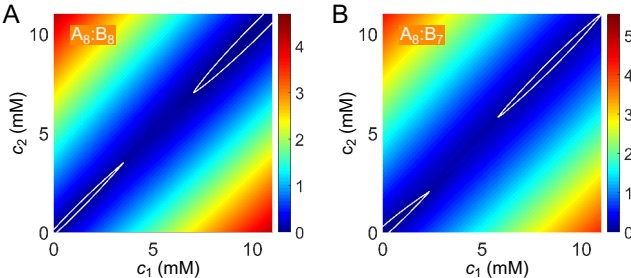

**Appendix 2—figure 2.** Free-energy density as a function of global sticker concentrations $(c_1, c_2)$ for (**A**) $A_8$:$B_8$ and (**B**) $A_8$:$B_7$ systems. White curves highlight the basins in the dilute dimer-dominated and dense gel-dominated regions.

To find the tie line going through a given initial concentration point $(c_1^{in}, c_2^{in})$, we adopt a modified vector method *Marcilla Gomis, 2011*. We first draw a line though this point along a direction defined by an angle $\alpha$, and then find the crossing points between this line and the phase boundaries, that is $(c_1^{dil}, c_2^{dil})$ and $(c_1^{den}, c_2^{den})$. The 'true' $\alpha$ is the one that minimizes the free-energy density of mixing $pF(c_1^{dil}, c_2^{dil}) + (1-p)F(c_1^{den}, c_2^{den})$, where $p = r^{den}/(r^{dil}+r^{den})$ is the volume fraction of dilute phase and $1 - p$ is the volume fraction of dense phase, and $r^{dil}$ and $r^{den}$ are the distances between the point $(c_1^{in}, c_2^{in})$ and the two points $(c_1^{dil}, c_2^{dil})$ and $(c_1^{den}, c_2^{den})$, respectively.

In order to compare with the simulated dilute- and dense-phase concentrations, we use the initial concentrations from simulations for the specified system at given stoichiometry $s$: $c_1 = c_t s/(1+s)$

and $c_2 = c_t/(1+s)$, where the total sticker concentration is $c_t$ = 6.64 mM. We then find the tie line going through this initial concentration point and the corresponding $(c_1^{\text{dil}}, c_2^{\text{dil}})$ and $(c_1^{\text{den}}, c_2^{\text{den}})$. For simplicity, we show the total dilute- and dense-phase concentrations $c^{\text{dil}} = c_1^{\text{dil}} + c_2^{\text{dil}}$ and $c^{\text{den}} = c_1^{\text{den}} + c_2^{\text{den}}$ in *Figure 6C and D*.

## A simple approximation for the free-energy density *F*

Finding the numerical solutions of *Equations (7) and (8)* becomes difficult with increasing valence. From the full solutions of $c_d$ and $c_b$ (*Appendix 2—figure 1*), we see that in the dimer-dominated region $c_b$ is almost 0, and in the independent bond-dominated region $c_d$ is almost 0. Therefore, we can approximate the free-energy density as the lower value of the two limiting cases

$$F = F_{\text{ni}} + \min(F_s^{\text{dim}}, F_s^{\text{ind}}) + F_{\text{ns}}, \tag{46}$$

where

$$\frac{F_s^{\text{dim}}}{k_B T} = \rho_d \ln K_d + \rho_d \ln \frac{\rho_d}{e} + (\rho_1 - \rho_d) \ln \frac{\rho_1 - \rho_d}{e} + (\rho_2 - \rho_d) \ln \frac{\rho_2 - \rho_d}{e} - \rho_2 \ln \frac{\rho_2}{e} - \rho_1 \ln \frac{\rho_1}{e}, \tag{47}$$

$$\frac{F_s^{\text{ind}}}{k_B T} = c_b \ln K_b + c_b \ln \frac{c_b}{e} + (c_1 - c_b) \ln \frac{c_1 - c_b}{e} + (c_2 - c_b) \ln \frac{c_2 - c_b}{e} - c_1 \ln \frac{c_1}{e} - c_2 \ln \frac{c_2}{e}, \tag{48}$$

and

$$\rho_d = \frac{1}{2}\left[ \rho_1 + \rho_2 + K_d - \sqrt{(\rho_1 + \rho_2 + K_d)^2 - 4\rho_1\rho_2} \right], \tag{49}$$

$$c_b = \frac{1}{2}\left[ c_1 + c_2 + K_b - \sqrt{(c_1 + c_2 + K_b)^2 - 4c_1 c_2} \right], \tag{50}$$

are solutions of *Equations (9) and (10)*. *Equation (46)* provides a very good approximation to the full expression for *F* (*Equation (13)*). The phase diagrams derived from *Equation (13) and (46)* are almost identical for $A_{14}{:}B_{14}$ (*Appendix 2—figure 3*). Results in *Appendix 2—figure 4* are obtained with this approximation (*Equations (46–50)*) as there are difficulties solving *Equations (7) and (8)* numerically for $A_{14}{:}B_{12\text{-}16}$ systems due to their high valences.

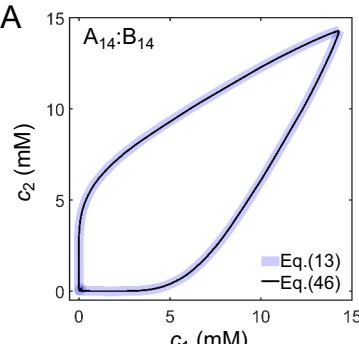 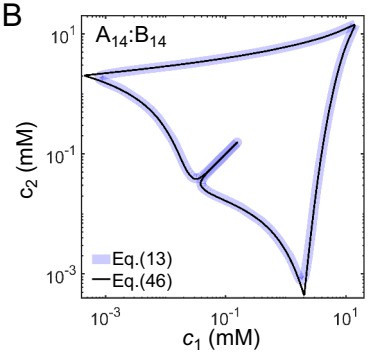

**Appendix 2—figure 3.** Comparison of phase diagrams derived from the full expression for *F* (*Equation (13)*) and the approximate expression (*Equation (46)*) shown on a (**A**) linear and (**B**) log scale for $A_{14}{:}B_{14}$ system.

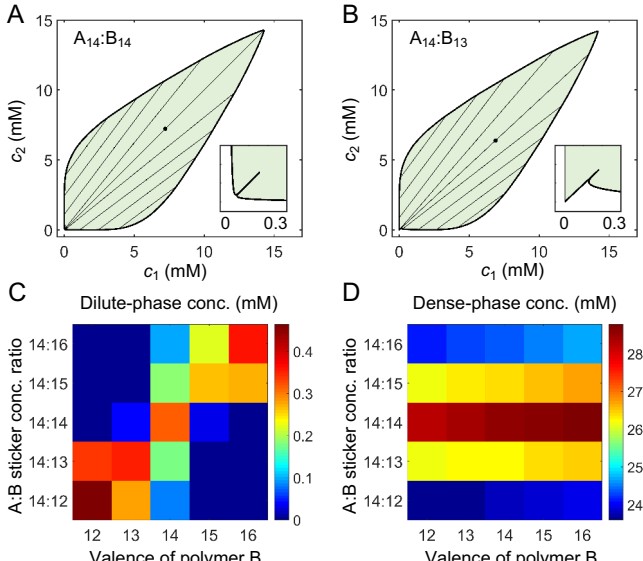

**Appendix 2—figure 4.** A dimer-gel theory predicts the magic-ratio effect. Phase diagrams of (**A**) $A_{14}:B_{14}$ and (**B**) $A_{14}:B_{13}$ systems: one-phase region white, two-phase region green. The dilute- and dense-phase concentrations are connected by representative tie lines. The tie line along the direction of equal polymer stoichiometry is denoted with a black dot. Inset: enlarged dilute-phase boundaries. Sticker concentrations in (**C**) dilute and (**D**) dense phases for systems $A_{14}:B_{12-16}$ at global sticker stoichiometries 14:12-16 and total sticker concentration 6.64 mM. Parameters: $v_b = 7 \times 10^{-2} \mathrm{mM}^{-1}$, $w_b = 5 \times 10^{-3} \mathrm{mM}^{-2}$, $K_b = 3.8 \times 10^{-3} \mathrm{mM}$, and $K_d$ in **Appendix 1—table 3**.

## Correlated binding in the dense phase

In the dimer-gel theory, we assume that stickers of different types can associate independently in the dense phase. However, as stickers belonging to the same polymer are tethered together, neighboring stickers in one polymer are more likely to bind to neighboring stickers in another polymer, that is there are correlations in binding. To quantify this correlation, we first identify consecutive segments in a polymer that bind to consecutive segments in another polymer. (For example, if in polymer 1 of type A, stickers 1, 2, 3, and 4 bind to stickers 2, 4, 3, and 5 of one polymer of type B, sticker 5 binds to sticker 8 of a second polymer of type B, and stickers 6, 7, and 8 bind to stickers 1, 2, and 3 of a third polymer of type B, then there are three individual segments in polymer 1 of type A with lengths 4, 1, and 3.) Clearly, what should be considered to be 'independent' are not individual stickers but rather these consecutively bound segments. To quantify the length of these segments, we measure the probability $p(l)$ that a bound sticker is in a segment of length $l$. **Appendix 2—figure 5** shows the probability distribution $p(l)$ for simulated $A_8:B_8$ and $A_{14}:B_{14}$ systems at equal stoichiometry. The mean length of 'independent' segments is 1.8 for both cases.

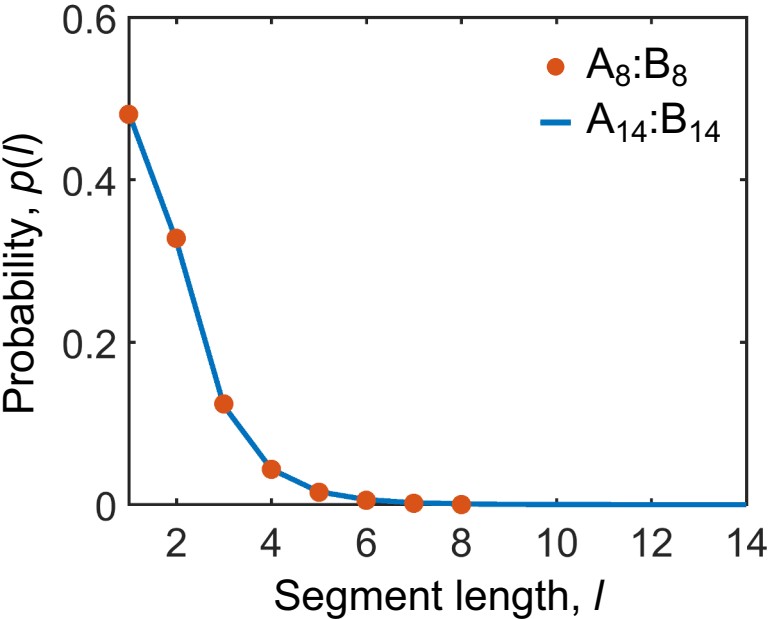

**Appendix 2—figure 5.** Binding between stickers of different types in the dense phase is correlated. Probability distribution $p(l)$ for finding a bound sticker to be in a consecutively bound segment of length $l$ in the dense phase of $A_8:B_8$ and $A_{14}:B_{14}$ systems at equal stoichiometry.

## Effects of model parameters on phase boundaries

The dimer-gel theory has only a handful of parameters: the valences $L_1$ and $L_2$ of polymers A and B, the dissociation constants $K_d$ and $K_b$ of dimers and independent bonds, and the nonspecific interaction parameters $v_b$ and $w_b$. These parameters together determine the competitiveness of the dilute dimer-phase with the dense gel-phase. We first fix $K_b = 3.8 \times 10^{-3}$mM, $v_b = 9 \times 10^{-2}$mM$^{-1}$, and $w_b = 7 \times 10^{-3}$mM$^{-2}$ (see previous section Determining Model Parameters for how these parameters are derived, and note that the values of $K_d$ are taken directly from simulations [*Appendix 1—table 3*]), and explore the dependence of phase boundaries on the valences $L_1$, $L_2$ and on the stoichiometry.

*Appendix 2–figure 6A and B* show phase diagrams and dilute-phase concentrations for $A_4:B_4$ to $A_{14}:B_{14}$ systems. For these equal valence systems, the dilute-/dense-phase concentrations decreases/increases with increasing valence, and the magic-ratio effect with respect to stoichiometry is enhanced with increasing valence in terms of dilute-phase peak-to-valley ratio. *Appendix 2–figure 6C and D* show phase diagrams and dilute-phase concentrations for $A_4:B_3$ to $A_{14}:B_{13}$ systems. Note that the shape of the dilute phase boundary transitions from a shoulder to a peak with increasing valence. All these features are consistent with the simulation results in *Figure 4*. *Appendix 2–figure 7B* shows the dilute-phase concentrations for $A_8:B_{6-10}$ systems at equal sticker stoichiometry. The dilute-phase concentration is sharply peaked at $A_8:B_8$, consistent with the simulation results in *Figure 3*.

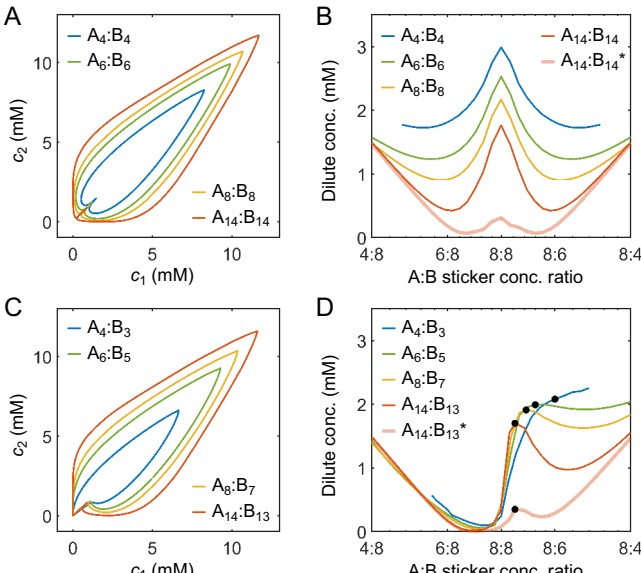

**Appendix 2—figure 6.** Effects of model parameters on the phase boundaries in the strong-binding regime. (**A**) Phase diagrams and (**B**) dilute-phase concentrations at different global stoichiometries for $A_4{:}B_4$ to $A_{14}{:}B_{14}$ systems. (**C**) Phase diagrams and (**D**) dilute-phase concentrations at different global stoichiometries for $A_4{:}B_3$ to $A_{14}{:}B_{13}$ systems. Black dots indicate equal *polymer* stoichiometries. Parameters: $K_b = 3.8 \times 10^{-3}$ mM, values of $K_d$ in *Appendix 1—table 3* for binding strength $U_0 = 14k_BT$, $v_b = 9 \times 10^{-2}$ mM$^{-1}$ and $w_b = 7 \times 10^{-3}$ mM$^{-2}$ for all systems except for $A_{14}{:}B_{14*}$ and $A_{14}{:}B_{13*}$ where $v_b = 7 \times 10^{-2}$ mM$^{-1}$ and $w_b = 5 \times 10^{-3}$ mM$^{-2}$. Total global sticker concentration 6.64 mM.

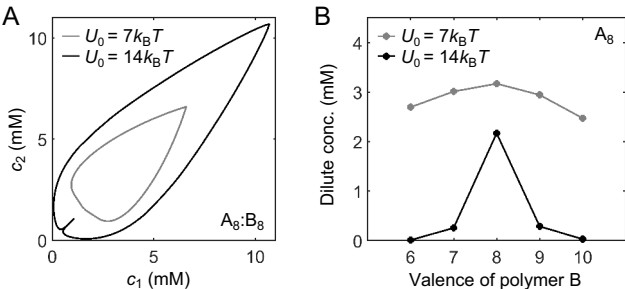

**Appendix 2—figure 7.** The magic-ratio effect disappears in the weak-binding regime. (**A**) Phase diagrams for $A_8{:}B_8$ system with different binding strengths $U_0 = 7k_BT$ (gray) and $U_0 = 14k_BT$ (black). (**B**) Dilute-phase concentrations for $A_8{:}B_{6\text{-}10}$ systems at equal global sticker stoichiometry with different binding strengths $U_0 = 7k_BT$ (gray) and $U_0 = 14k_BT$ (black). Parameters: $K_b = 1.88$ mM for $U_0 = 7k_BT$, $K_b = 3.8 \times 10^{-3}$ mM for $U_0 = 14k_BT$, values of $K_d$ in *Appendix 1—table 3*, $v_b = 9 \times 10^{-2}$ mM$^{-1}$, and $w_b = 7 \times 10^{-3}$ mM$^{-2}$. Total global sticker concentration 6.64 mM.

However, the dilute-phase concentration of $A_{14}{:}B_{14}$ does not quantitatively agree with simulation results (*Appendix 2–figure 6B* vs. *Figure 4A*). Also, the dilute-phase concentrations for unequal valence systems do not decrease with increasing valence (*Appendix 2–figure 6D* vs. *Figure 4C*). What is the origin of these discrepancies? Intuitively, the dense phase properties are determined by $K_b$, $v_b$, and, $w_b$. Using the same values of these parameters for systems with different valences means that we are treating the dense phases of these systems as exactly equivalent. However, there are more polymer backbone bonds in higher valence systems. These backbone bonds, from a mean-field point of view, act like attractive potentials between stickers, which effectively reduces the

nonspecific repulsion between stickers. Therefore, the dense phases of higher valence systems are energetically favored, and we expect correspondingly lower values of $v_b$ and $w_b$ for valence 14 systems compared to valence 8 systems. Indeed, we find that somewhat smaller nonspecific interaction parameters, $v_b = 7 \times 10^{-2} \mathrm{mM}^{-1}$ and $w_b = 5 \times 10^{-3} \mathrm{mM}^{-2}$, lead to quantitative agreement of the dilute- and dense-phase boundaries for $A_{14}{:}B_{14}$ and $A_{14}{:}B_{13}$ systems with the simulation results (*Appendix 2–figure 6B and D*, curves labeled with *, compared to *Figure 4A and C*). We thus use these parameters for $A_{14}{:}B_{12\text{-}16}$ systems in *Appendix 2–figure 4*.

Finally, the dimer-gel theory also predicts that the magic-ratio effect disappears in the weak-binding regime (*Appendix 2–figure 7A and B*, gray curves), consistent with the simulation results (*Figure 3A*).

