## [Decision Letter]

**Acceptance summary:**

In this work, Yaojun Zhang and coworkers develop a model based on concepts of polymer physics to study the formation of high density condensates of associating polymers. This occurs through a liquid-liquid phase separation process driven by the one-to-one association of specific stickers along the polymer chains. The authors provide an elaboration based on computations and a generalised theory to explain how a so-called magic number and / or magic ratio effect contributes to modulating the driving forces for phase transitions of multivalent proteins. In the case of two associating polymer species, the magic number effect occurs in the regime of strong specific interactions when the valence of one species is equal to or an integral multiple of the valence of the second species, allowing for the formation of small oligomers in the dilute phase. Here, the authors show that there is a magic ratio effect, whereby suppression of phase separation can be realized, again in the regime of strong, specific interactions, when the ratios of valencies are rational numbers. These findings, based on coarse-grained molecular dynamics simulations are recapitulated by a mean field theory and intuitively explained in terms of the interplay between conformational entropy and translational entropy costs. This work lays down conceptual foundations that will be helpful in understanding how signalling may be regulated by epigenetic changes to interactions among and stoichiometries of multivalent molecules.

**Decision letter after peer review:**

Thank you for submitting your article "Decoding the physical principles of two-component biomolecular phase separation" for consideration by *eLife*. Your article has been reviewed by two peer reviewers, and the evaluation has been overseen by a Reviewing Editor and Aleksandra Walczak as the Senior Editor. The following individual involved in review of your submission has agreed to reveal their identity: Rohit V Pappu (Reviewer #2).

The reviewers have discussed the reviews with one another and the Reviewing Editor has drafted this decision to help you prepare a revised submission. As explained below, both reviewers found your work important and valuable, as theory / computation have a lot to offer in terms of forging a conceptual understanding for the benefit of the biological soft matter community and a broader audience. However, there are a number of issues that need to be resolved and important comments that need to be addressed.

We would like to draw your attention to changes in our policy on revisions we have made in response to COVID-19 (https://elifesciences.org/articles/57162). Specifically, when editors judge that a submitted work as a whole belongs in *eLife* but that some conclusions require a modest amount of additional new data, as they do with your paper, we are asking that the manuscript be revised to either limit claims to those supported by data in hand, or to explicitly state that the relevant conclusions require additional supporting data.

Summary:

In this work, the authors provide an elaboration based on computations and a generalised theory to explain how a so-called magic number and / or magic ratio effect contributes to modulating the driving forces for phase transitions of multivalent proteins. In the case of two associating polymer species, the magic number effect occurs in the regime of strong specific interactions when the valence of one species is equal to or an integral multiple of the valence of the second species, allowing for the formation of small oligomers in the dilute phase. Here, the authors show that there is a magic ratio effect, whereby suppression of phase separation can be realized, again in the regime of strong, specific interactions, when the ratios of valencies are rational numbers. These findings, based on Langevin dynamics simulations are recapitulated by a mean filed theory and intuitively explained in terms of the interplay between conformational entropy and translational entropy costs. This work lays down conceptual foundations that will be helpful in understanding how signaling may be regulated by epigenetic changes to interactions among and stoichiometries of multivalent molecules.

Essential revisions:

The following list of comments is rather long, but I believe that most of them can be addressed without extensive additional work:

Regarding the presentation

1) The average reader of *eLife* is likely to become confused by the various terms used. Monomers are used somewhat interchangeably to refer to the specific modules within an associative polymer or to an individual associative polymer. I recommend the use of the term sticker or domain (the former is more precise) to refer to the individual modules within the associative polymers. Then, the authors can explain clearly that they are assessing the impact of: (a) varying the valence of stickers whilst keeping the global stoichiometry of stickers fixed; (b) varying the global stoichiometry (concentration) of stickers whilst keeping the valence fixed; and (c) for both (a) and (b) they are also titrating the strengths of inter-sticker interactions. A box defining the key terms and a figure showing exactly what is being titrated and how would be very helpful to the authors' cause.

2) The main important features of Figure 4A and B should be explained more clearly. In particular, Figure 4B is not discussed in the text.

Regarding the model

3) Spacers are not likely to be passive. This point has been made in a few papers that deploy the stickers-and-spacers formalism. Please see: https://elifesciences.org/articles/30294. Accordingly, there will be an interplay between the effects of spacers and the triad of effects due to stickers. The effects of are largely ignored. It would be important to highlight this fact because even for the two-component system that motivates the current work, there are known effects due to spacers that cannot be ascribed to being non-specific effects. Please see: http://www.pnas.org/content/112/47/E6426.long.

4) Specific, inter-sticker interactions should be anisotropic (please see Choi, Holehouse and Pappu, 2020a). In the model used for the simulations, the inter-sticker interactions are isotropic. To work around this problem, the potential includes a repulsive term between stickers of the same type. This creates a conundrum because it introduces auxiliary energy and length scales. Therefore, the effective heterotypic inter-sticker interaction energies will be different from what is prescribed by the values for *U*_0_. Of additional importance is the diluting effect of the homotypic repulsive interactions. This could have non-trivial many-body consequences and further, it will impact the cluster size distributions, and the pressure. It would help if the authors were to furnish two details from analysis of their simulations: How does the internal pressure scale with the energy and length scales associated with repulsive homotypic interactions? And how do these repulsive interactions impact the *U*_0_ dependence of the magic ratio effect? One set of calculations should suffice.

Regarding the nature of the transition

5) The dimer gel theory is very interesting. Following Harmon et al., 2017 and 2018, Choi et al., 2019, and the seminal work of Semenov and Rubinstein, 1998, it is important to recognize that phase transitions in associative polymeric systems are best thought of as phase separation aided percolation. Accordingly, it would help to calculate the percolation threshold from the simulations and from the theory as well. For the computational work, this will help us understand how *U*_0_, the magic ratio, and valence impact the interplay between density transitions (aka phase separation) and percolation. Furthermore, there are two new generalizations of the work of Semenov and Rubinstein that are relevant to the theoretical work here. Please see: https://doi.org/10.1021/acs.macromol.8b00661 and http://dx.doi.org/10.1103/PhysRevE.102.042403. A comparison of these generalizations and the dimer gel theory introduced here would be very helpful and insightful, especially in the context of bond cooperativity.

6) Throughout, but most notable in the subsection “Dimer-gel Theory Results”, there seems to be confusion of phase separation (resulting from a thermodynamic instability) and gelation (a connectivity transition). The two cases need to be delineated more clearly.

7) Are the phase boundaries (both theoretical and from simulations) spinodals or binodals?

Regarding the simulations

8) The MD protocol is unusual. Why include harmonic bonding potentials (instead of FENE)? Why use a softened LJ potential with an attractive tail? Typically Monte Carlo moves are needed to enforce 1:1 binding (pioneered by Arlette Baljon) or a carefully-designed combination of 2-body and 3-body potentials (pioneered by Francesco Sciortino). The model in the current manuscript may also succeed at enforcing 1:1 binding, but this needs to be proven and shown not to introduce other biases.

9) The phase separation analysis of the MD results could be explained more rigorously. What checks have been done to confirm phase separation rather than a sol-gel transition or other artifact of simulation preparation. Have you confirmed that the two resultant phases are in chemical and mechanical equilibrium? Do the simulation pressure or structure factor show signatures of phase separation?

Regarding the Results and Discussion

10) The mean-field assumption allows for useful analytical results, but is not expected to be valid for the types of systems simulated and discussed. The chains of such short length are not overlapping over most of the concentration regime discussed and locations of stickers is highly correlated instead of mean-field. Further, as would be expected, simulations by C. E. Sing. and A. Alexander-Katz (Macromolecules 2011) have shown sticker-spacer descriptions do not extend to the case where every bead is sticky.

11) A primary limitation of the work is the very short chains used in the MD simulations and necessary for the results to be useful. In a true polymeric limit and for most real systems, the numbers of beads and stickers are an order of magnitude larger than discussed in the paper. At this large N limit, this magic ratio effect becomes rather trivial. The extent to which the results presented are a feature of the discrete limit of very short oligomers should be properly discussed.

12) The results in Figure 3 raise some questions. The results in the left column show a different non-monotonic dependence of the dilute phase concentration on the A:B monomer concentration ratio (best referred to as the concentration ratio of A : B stickers) when compared to the dense phase concentration. First, given that there are two species, how are concentrations in the dilute and dense phases being reduced to a single number? Second, there ends up being a regime where the dilute phase concentration of stickers increases and yet the dense phase concentration of stickers also increases, leading to what seems like a conservation of the width of the two-phase regime. Likewise, the opposite scenario obtains in a different regime, wherein the dilute phase concentration decreases and over on the right column, we see a decrease in the dense phase concentration. Does this imply that the width of the two-phase regime, quantifiable as the log10(c_dense_/c_dilute_) stays fixed as the A:B sticker concentration ratio changes? This aspect needs some thinking and further analysis or clarification. I am missing something because the inference I draw from the increase in the concentration of the dilute arm is of weakening the driving forces for phase separation. However, if that is accompanied by an increase in the concentration of the dense phase, then it would imply that despite the apparent diminution in the driving forces for phase separation, an increase in the concentration of the dense phase results in equalizing the chemical potential with the dilute phase. The converse also seems to obtain from the results. And this is perplexing.

13) The dense and dilute phase concentrations never seem to be more than an order of magnitude apart. This too is perplexing. Additionally, the concentrations, even in the dilute phase are in the millimolar range. And there is a perplexing conversion of an interaction energy of 9kT to dissociation constants, which ends up being rather high, in the millimolar range. This is confusing because the authors propose that these estimates are on a par with SUMO-SIM and SHR-PRM interactions, which is not the case as these interactions are in the low μM rather than millimolar range. There appears to an issue here with converting units into molar scales. I recommend the usage of volume fractions. For polymeric systems, volume fractions are the most robust route for quantifying concentrations – a point emphasized by Flory back in the early days of polymer science.

14) Finally, given the audience, i.e., readers of *eLife*, the nagging concern will pertain to the regime and systems where the magic number and magic ratio effects are manifest. This might be relevant and / or applicable in engineered systems. This is alluded to in the Discussion. It might also be the case that given the rather strong interactions required to observe the magic number and / or magic ratio effect, this might not be realizable in vivo. The challenge at high interaction strengths will be problems with solubility, the formation of precipitates and / or the onset of the glass transition. In fact, the latter has been speculated as being important in biology, see https://doi.org/10.1016/j.sbi.2016.05.002 and https://elifesciences.org/articles/09347. Indeed, the simulations of Choi et al., 2019, show this onset of glassy behavior as interaction strengths increase or temperature is lowered. Might it be the case that the magic number or ratio effects are masked by the drive to avoid precipitation and / or glassy behavior? Alternatively, as the authors suggest, the magic number / ratio effect might be a way to regulate phase behavior. Some elaboration of these issues, beyond what is currently offered, might be beneficial to the reader, especially if can be accompanied by pointers to specific systems that motivate experimental investigations.

---

## [Author Response]

Essential revisions:The following list of comments is rather long, but I believe that most of them can be addressed without extensive additional work:Regarding the presentation1) The average reader of eLife is likely to become confused by the various terms used. Monomers are used somewhat interchangeably to refer to the specific modules within an associative polymer or to an individual associative polymer. I recommend the use of the term sticker or domain (the former is more precise) to refer to the individual modules within the associative polymers. Then, the authors can explain clearly that they are assessing the impact of: (a) varying the valence of stickers whilst keeping the global stoichiometry of stickers fixed; (b) varying the global stoichiometry (concentration) of stickers whilst keeping the valence fixed; and (c) for both (a) and (b) they are also titrating the strengths of inter-sticker interactions. A box defining the key terms and a figure showing exactly what is being titrated and how would be very helpful to the authors' cause.

We agree with the reviewers that it is more suitable to use stickers/domains to refer to specific modules within the associative polymers for the readers of *eLife*, and have made corresponding changes throughout the manuscript. We have also added Figure 1 along with its caption to serve as a guideline of our work.

2) The main important features of Figure 4A and B should be explained more clearly. In particular, Figure 4B is not discussed in the text.

We agree – we have added more discussion concerning Figure 4 in the text:

“As for the dense phase concentration, it decreases monotonically as the global sticker stoichiometry departs from 1 and as the valence of polymers decreases (Figure 5B). This again indicates that the anomalous dependence of the dilute-phase concentration on valence and stoichiometry does not arise from special properties of the dense phase.”

Regarding the model3) Spacers are not likely to be passive. This point has been made in a few papers that deploy the stickers-and-spacers formalism. Please see: https://elifesciences.org/articles/30294. Accordingly, there will be an interplay between the effects of spacers and the triad of effects due to stickers. The effects of are largely ignored. It would be important to highlight this fact because even for the two-component system that motivates the current work, there are known effects due to spacers that cannot be ascribed to being non-specific effects. Please see: http://www.pnas.org/content/112/47/E6426.long.

We thank the reviewers for highlighting important recent work on the role of spacers in phase separation. To address this and the following point regarding the length scale of homotypic repulsion, we have added the following sentences in the Discussion:

“Furthermore, real biological systems are more complex than our simple model. For example, there can be multiple-to-one binding, multiple components, and the spacers/linkers can also play nontrivial roles Banjade et al., 2015; Harmon et al., 2017.”

“On the other hand, the dense-phase concentration strongly depends on the steric repulsion – increasing the sticker size from 2.5nm to 2.9nm decreases the dense phase concentration by a factor of 2 (Appendix 1—figure 5B). This is consistent with results from previous studies on the role of linkers: a self-avoiding random coil linker which occupies a large volume can substantially lower the dense-phase concentration and even prevent phase separation Harmon et al., 2017.”

4) Specific, inter-sticker interactions should be anisotropic (please see Choi et al., 2020a). In the model used for the simulations, the inter-sticker interactions are isotropic. To work around this problem, the potential includes a repulsive term between stickers of the same type. This creates a conundrum because it introduces auxiliary energy and length scales. Therefore, the effective heterotypic inter-sticker interaction energies will be different from what is prescribed by the values for U_0_. Of additional importance is the diluting effect of the homotypic repulsive interactions. This could have non-trivial many-body consequences and further, it will impact the cluster size distributions, and the pressure. It would help if the authors were to furnish two details from analysis of their simulations: How does the internal pressure scale with the energy and length scales associated with repulsive homotypic interactions? And how do these repulsive interactions impact the U_0_ dependence of the magic ratio effect? One set of calculations should suffice.Ur(r)=4ϵ[(σr)12−(σr)6+14], r<21/6σ,

We agree with the reviewers that biological sticker-sticker interactions are anisotropic. It is this anisotropy and volume exclusion between stickers that lead to one-to-one association. The model we developed is highly simplified, keeping the key property (one-to-one association) but removing system specific details. It is therefore a conceptual model. That said, our approach does not introduce additional energy or length scales: the repulsive interactions between like stickers correspond to steric repulsion which must be present in any model, and the energy scale of this repulsion is essentially irrelevant, since the repulsive interactions are effectively in the regime of “hard-sphere” repulsion. Quantitative descriptions of real systems will likely require additional features, for example using sticky patches to model anisotropic binding. To address the reviewers’ question about the dependence on the length scale of repulsion, we performed new simulations for the A8:B8 system. Here, we used the standard repulsive LJ potential for interactions between stickers of the same typewhere ϵ=1kBT. To examine the role of the repulsive length scale, we employed two choices of sticker diameters σ = 2.9 nm (second virial coefficient matched to the soft LJ potential used in the manuscript) and a smaller bead σ = 2.5 nm. The bond potential and the attraction between stickers of different types are the same as previously employed (i.e., Equations 16 and 18 in the manuscript), except we used binding parameters r_0_ = 1.5 nm and U0=12kBT which lead to weaker binding. The dilute- and dense-phase sticker concentrations and dense-phase volume fraction for bead diameter σ = 2.9 nm are shown in Author response image 1, and for σ = 2.5 nm are shown in Author response image 2.

**Author response image 2. respfig2:** 

(Note that the overlapping volume between two bound stickers is counted once in the volume fraction calculation, i.e., two perfectly overlapping stickers would occupy a volume of one sticker.) Note also that because we used a harmonic potential for backbone linker bonds which has a well-defined mean bond length 4.7 nm (not sensitive to the choice of σ), the changes in the dilute- and dense-phase concentrations purely reflect the effect of the homotypic repulsive interactions. (We show in point 13 below how these quantities depend on linker length.)Comparing the two cases, the reviewers are certainly correct that there is a diluting effect of the homotypic repulsive interactions. A stronger repulsion/larger repulsive length scale results in a lower concentration and volume fraction in the dense phase. The onset binding strength U0 for the magic-number/magic-ratio effect also becomes higher. For σ = 2.9 nm there is almost no magic-ratio effect at U0=12kBT and the effect is there at U0=14kBT (similar to what is shown in Figure 3 in the manuscript), whereas for σ = 2.5 nm the effect is already quite significant at U0=12kBT.

Related to this effect, if we were to model the linkers explicitly and if the linkers fall into the category of self-avoiding random coil (SARC) (Harmon et al., 2017), then, qualitatively, compared to a Flory random coil (FRC) linker, the SARC linker effect can be viewed as increasing the repulsive length scale of stickers (assume the SARC and FRC linkers have the same mean end-to-end distance), and therefore could lead to weakened magic-number/magic-ratio effects.

Please see the additional remarks we have added to the Discussion noted in our response to point 3.

Regarding the nature of the transition5) The dimer gel theory is very interesting. Following Harmon et al., 2017 and 2018, Choi et al., 2019, and the seminal work of Semenov and Rubinstein, 1998, it is important to recognize that phase transitions in associative polymeric systems are best thought of as phase separation aided percolation. Accordingly, it would help to calculate the percolation threshold from the simulations and from the theory as well. For the computational work, this will help us understand how U_0_, the magic ratio, and valence impact the interplay between density transitions (aka phase separation) and percolation. Furthermore, there are two new generalizations of the work of Semenov and Rubinstein that are relevant to the theoretical work here. Please see: https://doi.org/10.1021/acs.macromol.8b00661 and http://dx.doi.org/10.1103/PhysRevE.102.042403. A comparison of these generalizations and the dimer gel theory introduced here would be very helpful and insightful, especially in the context of bond cooperativity.

We agree with the reviewers that the dense phase of associative polymers is a percolating network, and we thank the reviewer for pointing us to relevant theories. To our knowledge, and as argued by Semenov and Rubenstein, a bulk percolation threshold (aka homogeneous gelation transition) only exists at high temperatures/weak binding where phase separation doesn’t happen. Our current study, however, is focused on the low temperature regime. At these low temperatures, while we expect the percolation threshold cp to lie between the dilute- and dense-phase concentrations, if we were to simulate a system at cp, the system would automatically phase separate. That said, following the reviewers’ suggestion we can gain some insight into a characteristic density for homogeneous percolation in our model by characterizing the percolation threshold in the weak binding regime U0=8kBT, which avoids phase separation but also prevents many-to-one binding. To present these results, we have incorporated a new section and figure in Appendix 1 (subsection “Determining the percolation threshold”, Appendix 1—figure 3).

We have also highlighted the relevant theories that generalize the work of Semenov and Rubenstein in the Discussion:

“One of the major assumptions of the dimer-gel theory is a mean-field approximation. […] This approximation captures a key feature of the dense phase, namely that a single polymer binds to multiple partners.”

6) Throughout, but most notable in the subsection “Dimer-gel Theory Results”, there seems to be confusion of phase separation (resulting from a thermodynamic instability) and gelation (a connectivity transition). The two cases need to be delineated more clearly.

Because the dense phase of associative polymers is a gel, we used “gel” interchangeably with “dense phase” in some cases. To avoid possible confusion, we have made the following changes:

“However, cluster size may reflect a sol-gel percolation transition rather than a thermodynamic phase transition Harmon et al., 2017, and thus provides at best a qualitative measure of phase separation.”

“Comparing the two expressions, dimers are favored at low concentrations, whereas a network of independent bonds is favored at high concentrations.”

7) Are the phase boundaries (both theoretical and from simulations) spinodals or binodals?

The phase boundaries are binodals, and we have now clarified this point: “To find the binodal phase boundaries, we simulate hundreds of polymers of types A and B…” and “Finally, to extract the binodal phase boundaries, we substitute…”

Regarding the simulations8) The MD protocol is unusual. Why include harmonic bonding potentials (instead of FENE)? Why use a softened LJ potential with an attractive tail?Ur(r)=4ϵ[(σr)12−(σr)6+14], r<21/6σ.Ub(r)=−12KR02ln⁡[1−(r−σR0)2], r<σ+R0.Ua(r)=−12U0(1+cos⁡πrr0), r<r0,

We agree with the reviewers that our MD potentials are not the most standard choices. Part of our reason for using a harmonic potential and a softened LJ potential was to speed up the MD simulations, as the softened potentials allow a larger integration time step. To demonstrate the robustness of our conclusions to the choice of MD potentials, we have performed new simulations for the A8:B8 system using FENE and standard LJ potentials. Briefly, the interaction between stickers of the same type is purely repulsive We take ϵ=1kBT and σ=3 nm, which results in a repulsive potential comparable to our previous softened LJ potential. Neighboring stickers are connected by a FENE potentialWe take K=0.3kBT/nm2 and R0=7 nm, which (together with the repulsive interaction) results in the same mean bond length 4.7 nm as for our previous harmonic potential. (This potential resembles a 20 amino acid disordered FCS linker of persistence length 0.8 nm.) The attractive interaction between different types of stickers has the same form as in our manuscript: where r0=1.5 nm is the cutoff distance. For U0=14kBT, we show the total dilute- and dense-phase sticker concentrations in Author response image 3.

**Author response image 3. respfig3:** 

The results are qualitatively unchanged from the results presented in Figure 3 in our manuscript. Quantitatively, the dilute-phase concentrations drop by a factor of 2 and the dense-phase concentrations are comparable to the reported values. We conclude that our conceptual results on the magic-number/magic-ratio effects are robust to the detailed choice of potentials.

Typically Monte Carlo moves are needed to enforce 1:1 binding (pioneered by Arlette Baljon) or a carefully-designed combination of 2-body and 3-body potentials (pioneered by Francesco Sciortino). The model in the current manuscript may also succeed at enforcing 1:1 binding, but this needs to be proven and shown not to introduce other biases.

Testing for biases is certainly important. We always checked the number of bound partners for all the stickers in our simulations. A sticker is considered to be a partner of another sticker if the two are of different types and are within the cutoff distance r0 of the attractive potential. Across our simulations, the average fraction of stickers that have more than one partner is less than 0.001%.

Regarding other possible biases, we note that Langevin dynamics can be viewed as moves to achieve thermodynamic equilibrium. We carefully designed the initial state of the system, and ran long enough simulations for the system to fully equilibrate, and are thus confident that there is no bias against equilibrium.

One possible “bias” is that stickers overlap upon binding, which leads to a reduction in volume. In principle, this could enhance the effect of “crowding promoted binding” in the dense phase, i.e., binding being further favored by the resulting reduction in volume. However, the volume fraction of the dense phase, ∼10%, is not high, so this effect is very modest.

9) The phase separation analysis of the MD results could be explained more rigorously. What checks have been done to confirm phase separation rather than a sol-gel transition or other artifact of simulation preparation. Have you confirmed that the two resultant phases are in chemical and mechanical equilibrium? Do the simulation pressure or structure factor show signatures of phase separation?

Phase separation and a sol-gel transition may be ambiguous in the high temperature regime and/or close to a critical point. In our case, we are working in the strongly condensed regime, i.e., low temperature and adequate polymer concentrations, and have excluded systems where the dilute- and dense-phase concentrations become comparable (e.g., A8:B8 system at A:B sticker ratio 8:6). For all the systems reported in the manuscript, there is a clearly-defined, large dense-phase cluster located in a small region of the simulation box and a dilute-phase of polymers/oligomers dispersed throughout the rest of the box (representative snapshots are shown in Figure 2C and 2D), it is thus clear these systems are phase separating (and thus undergoing dense-phase gelation driven by phase separation).

Langevin dynamics ensures a Boltzmann distribution. We have carefully designed the initial state of the system (a fully connected dense phase) to prevent the system becoming stuck in a local minimum (such as dispersed dense droplets that take a long time to merge). We have then run a long simulation to allow the dense phase to equilibrate with the dilute phase. The relaxation time of the system depends on the sticker-sticker bond lifetime τ. To ensure that the dilute and dense phases are in equilibrium, all simulations were equilibrated ∼2000τ before data was recorded. We also checked whether there are systematic deviations between the first and second halves of the recorded simulation, and find consistent results between the two halves.

We have added a few sentences to make the discussion of phase separation more rigorous:

“Across our simulations, on average the fraction of stickers that have more than one partner is less than 0.001%.” and “The relaxation time of the system depends on the sticker-sticker bond lifetime; to ensure that the dilute and dense phases are in equilibrium…”

Regarding the Results and Discussion10) The mean-field assumption allows for useful analytical results, but is not expected to be valid for the types of systems simulated and discussed. The chains of such short length are not overlapping over most of the concentration regime discussed and locations of stickers is highly correlated instead of mean-field. Further, as would be expected, simulations by C. E. Sing. and A. Alexander-Katz (Macromolecules 2011) have shown sticker-spacer descriptions do not extend to the case where every bead is sticky.

We agree with the reviewers that the mean-field assumption is not valid over the entire range of possible concentrations as the chains may not overlap. However, to locate the dilute and dense-phase boundaries with our convex-hull analysis, we only need our dense-phase mean-field approximation to be valid in the concentration regime comparable to the dense-phase concentrations. The mean-field approximation is still not fully quantitative at dense-phase concentrations as we found correlations in binding in our simulations (correlation length ~1.8 beads) that go beyond mean-field theory. Therefore, a simulated A14:B14 system has an effective valence ~8 and therefore behaves more like an A8:B8 system in theory. Nevertheless, the theory works better for systems with longer linker lengths (or smaller beads), where binding is less correlated in the dense phase.

11) A primary limitation of the work is the very short chains used in the MD simulations and necessary for the results to be useful. In a true polymeric limit and for most real systems, the numbers of beads and stickers are an order of magnitude larger than discussed in the paper. At this large N limit, this magic ratio effect becomes rather trivial. The extent to which the results presented are a feature of the discrete limit of very short oligomers should be properly discussed.

Our work focuses on proteins with complementary domains (“stickers”) instead of intrinsically disordered proteins which could have very large numbers of effective stickers. Examples of the kinds of systems that motivated our work include SUMO-SIM (Banani et al., 2016), SHR-PRM (Li et al., 2012), and Rubisco-EPYC1 (Rosenzweig et al., 2017), which are all short in the sense of having <10-15 stickers per polymer. That said, the magic-number/magic-ratio effects extend to large *N* and, in fact, the effects become stronger with increasing polymer valences (Figure 3 in the manuscript).

12) The results in Figure 3 raise some questions. The results in the left column show a different non-monotonic dependence of the dilute phase concentration on the A:B monomer concentration ratio (best referred to as the concentration ratio of A : B stickers) when compared to the dense phase concentration. First, given that there are two species, how are concentrations in the dilute and dense phases being reduced to a single number?

The concentrations shown in Figure 3 are the sum of the A and B sticker concentrations, which we now refer to as the total sticker concentration.

Second, there ends up being a regime where the dilute phase concentration of stickers increases and yet the dense phase concentration of stickers also increases, leading to what seems like a conservation of the width of the two-phase regime. Likewise, the opposite scenario obtains in a different regime, wherein the dilute phase concentration decreases and over on the right column, we see a decrease in the dense phase concentration. Does this imply that the width of the two-phase regime, quantifiable as the log10(c_dense_/c_dilute_) stays fixed as the A:B sticker concentration ratio changes? This aspect needs some thinking and further analysis or clarification.

These are certainly correct statements. In fact, the nonmonotonic dependence of the dilute-phase concentration on A:B sticker ratio is a hallmark of the magic-number/magic-ratio effect. Taking the simulation in point 8 as an example, c_dense_/c_dilute_ vs. A:B sticker concentration ratio (plotted on a log-log scale) first increases and then decreases as the A:B ratio deviates from 1. This nonmonotonicity is entirely due to the magic-number/magic-ratio effect raising the dilute-phase concentration at equal A:B ratio.

**Author response image 4. respfig4:** 

I am missing something because the inference I draw from the increase in the concentration of the dilute arm is of weakening the driving forces for phase separation. However, if that is accompanied by an increase in the concentration of the dense phase, then it would imply that despite the apparent diminution in the driving forces for phase separation, an increase in the concentration of the dense phase results in equalizing the chemical potential with the dilute phase. The converse also seems to obtain from the results. And this is perplexing.

For conventional polymer phase separation (such as systems described by Flory-Huggins mean-field theory), the increase of the dilute-phase concentration is connected to a weakening of the driving forces for phase separation and hence a lowered dense-phase concentration. However, this is not always the case for more complicated systems, in particular when phase separation is not simply driven by a competition between energy and entropy. For our two-component system, phase separation is essentially driven by a competition between translational and conformational entropy. Briefly, as the A:B sticker ratio approaches 1, the dilute-phase concentration increases. This means that the chemical potential of dimers in the dilute phase increases. As the dense phase is in equilibrium with the dilute phase, the chemical potential in the dense phase necessarily also increases. Mechanistically, why does the dense phase have a higher concentration and a higher chemical potential at A:B ratio 1? In fact, this is only true in the strong binding regime. As all the stickers have to be paired when binding is strong at A:B ratio 1, this leads to a contraction of the dense phase, and thus a high concentration. This same contraction leads to conformational frustration in the dense phase, reduces the conformational entropy, and thus raises the chemical potential.

13) The dense and dilute phase concentrations never seem to be more than an order of magnitude apart. This too is perplexing. Additionally, the concentrations, even in the dilute phase are in the millimolar range.

We agree with the reviewers that our dense- and dilute-phase concentration ratios as well as the dilute-phase concentrations are not in the range typical of most of biological systems. We have explored how these quantities depend on the simulation parameters. In particular, we found that the dilute-phase concentration is very sensitive to the linker length. Taking U0=12kBT,K=0.3kBT/nm2, and R0=7nm (a 20 amino acid FRC linker of persistence length 0.8 nm, mean bond length 4.7 nm), all other parameters are the same as listed in point 8, see Author response image 5, where as U0=12kBT,K=0.15kBT/nm2, and R0=14 nm (a 40 amino acid FRC linker of persistence length 0.8 nm, mean bond length 5.9nm) yields are shown in Author response image 6

**Author response image 5. respfig5:** 

**Author response image 6. respfig6:** 

Thus, upon increasing the linker length ~25%, the dilute-phase concentration drops more than a factor of 10, while the dense-phase concentration remains almost unchanged. Therefore, higher dense-phase to dilute-phase concentration ratios and lower dilute-phase concentrations are achievable with minimal modifications. We acknowledge that many parameter choices in the manuscript are at least in part motivated by promoting simulation speed. We have incorporated the above results in our manuscript “Reducing the steric repulsion between beads of the same type has a similar effect (Appendix 1—figure 5A). More significantly, increasing the mean linker length from 4.7 nm to 5.9 nm leads to a more than 10-fold reduction in the dilute-phase concentration (Appendix 1—figure 6A)…” and a new section Effects of nonspecific interactions and linker length in Appendix 1.

And there is a perplexing conversion of an interaction energy of 9kT to dissociation constants, which ends up being rather high, in the millimolar range. This is confusing because the authors propose that these estimates are on a par with SUMO-SIM and SHR-PRM interactions, which is not the case as these interactions are in the low μM rather than millimolar range.Kd=[∫0r04πr2exp[−Ua(r)kBT]dr]−1,

We designed our potentials to fall in the range of typical sticker-sticker Kd values. The sticker-sticker dissociation constant can be found fromwhere r0 is the cutoff distance for an attractive potential. U0 is the minimum of Ua(r). Importantly, Kd is not only determined by U0 but also by the volume of the binding pocket. We found Kd=0.4mM for U0=9kBT and r0=2nm based on the above expression. The reported sticker-sticker dissociation constant for SUMO-SIM is Kd=0.01mM (Banani et al., 2016) and for SHR-PRM is Kd=0.35mM (Li et al., 2012). The magic-number/magic-ratio effects are enhanced with increasing binding strength. We find that Kd≲0.4mM provides a rough threshold for the onset of these effects, although this threshold also depends on valence of the polymers, sticker size, and bond length, etc.

There appears to an issue here with converting units into molar scales. I recommend the usage of volume fractions. For polymeric systems, volume fractions are the most robust route for quantifying concentrations – a point emphasized by Flory back in the early days of polymer science.

We agree with the reviewers that volume fraction is a sensible quantity for characterizing dense-phase concentration, the dense-phase volume fractions are ~10% in the simulations in our manuscript. We have accordingly added this quantification in the text where it will be most helpful. “We also report in Appendix 1—figure 5C the volume fraction of the polymers in the dense phase, which is ~10%, comparable to the volume fraction of proteins in the cell cytoplasm.”

14) Finally, given the audience, i.e., readers of eLife, the nagging concern will pertain to the regime and systems where the magic number and magic ratio effects are manifest. This might be relevant and / or applicable in engineered systems. This is alluded to in the Discussion. It might also be the case that given the rather strong interactions required to observe the magic number and / or magic ratio effect, this might not be realizable in vivo. The challenge at high interaction strengths will be problems with solubility, the formation of precipitates and / or the onset of the glass transition. In fact, the latter has been speculated as being important in biology, see https://doi.org/10.1016/j.sbi.2016.05.002 and https://elifesciences.org/articles/09347. Indeed, the simulations of Choi et al., 2019, show this onset of glassy behavior as interaction strengths increase or temperature is lowered.

We share the same uncertainty as the reviewers whether the magic-number/magic-ratio effects can manifest and be utilized by cells as a way to regulate condensates. From a purely theoretical point of the view, as long as the sticker-sticker dissociation constant is below a certain (submillimolar) threshold, there is a very real chance to observe these effects. In this sense, many condensate scaffold components satisfy this criterion. However, real biological systems are of course far more complex than our simple model. There can be multiple-to-one binding, multiple components, and, as has been pointed out by the reviewers, the spacers can also play important, but complicated roles. So the in vivo relevance of the effects explored here remains an open question.

Might it be the case that the magic number or ratio effects are masked by the drive to avoid precipitation and / or glassy behavior? Alternatively, as the authors suggest, the magic number / ratio effect might be a way to regulate phase behavior. Some elaboration of these issues, beyond what is currently offered, might be beneficial to the reader, especially if can be accompanied by pointers to specific systems that motivate experimental investigations.

We thank the reviewers for sharing these interesting thoughts, and we have modified the text accordingly “However, the magic-ratio effect has not been observed in these systems Li et al., 2012; Banani et al., 2016, possibly due to size and linker length mismatch between the two associating polymers.[…] Currently, the in vivo relevance of the effects explored in this work remains an open question.”